# Microbial species and intraspecies units exist and are maintained by ecological cohesiveness coupled to high homologous recombination

Roth E. Conrad [1,6], Catherine E. Brink[1,6], Tomeu Viver[2,3], Luis M. Rodriguez-R[4], Borja Aldeguer-Riquelme [1], Janet K. Hatt [1], Stephanus N. Venter[5], Ramon Rossello-Mora [2,7] ✉, Rudolf Amann [3,7] ✉ & Konstantinos T. Konstantinidis [1,7] ✉

Recent genomic analyses have revealed that microbial communities are predominantly composed of persistent, sequence-discrete species and intraspecies units (genomovars), but the mechanisms that create and maintain these units remain unclear. By analyzing closely-related isolate genomes from the same or related samples and identifying recent recombination events using a novel bioinformatics methodology, we show that high ecological cohesiveness coupled to frequent-enough and unbiased (i.e., not selection-driven) horizontal gene flow, mediated by homologous recombination, often underlie these diversity patterns. Ecological cohesiveness was inferred based on greater similarity in temporal abundance patterns of genomes of the same vs. different units, and recombination was shown to affect all sizable segments of the genome (i.e., be genome-wide) and have two times or greater impact on sequence evolution than point mutations. These results were observed in both *Salinibacter ruber*, an environmental halophilic organism, and *Escherichia coli*, the model gut-associated organism and an opportunistic pathogen, indicating that they may be more broadly applicable to the microbial world. Therefore, our results represent a departure compared to previous models of microbial speciation that invoke either ecology or recombination, but not necessarily their synergistic effect, and answer an important question for microbiology: what a species and a subspecies are.

Whether species exist and, if so, how to recognize them are challenging questions to answer for many microbes including Bacteria and Archaea (the prokaryotes), with obvious practical implications for identifying or regulating organisms of clinical or environmental importance[1–4]. Recent large-scale surveys of prokaryotic communities (metagenomes) as well as isolate genomes have revealed that their diversity is predominantly organized in sequence-discrete clusters or units that may be equated to species. Specifically, genomes of the

[1]Georgia Institute of Technology, Atlanta, GA, USA. [2]Mediterranean Institutes for Advanced Studies (IMEDEA, CSIC-UIB), Esporles, Spain. [3]Max Planck Institute for Marine Microbiology, Bremen, Germany. [4]University of Innsbruck, Innsbruck, Austria. [5]University of Pretoria, Pretoria, South Africa. [6]These authors contributed equally: Roth E. Conrad, Catherine E. Brink. [7]These authors jointly supervised this work: Ramon Rossello-Mora, Rudolf Amann, Konstantinos T. Konstantinidis. ✉e-mail: ramon@imedea.uib-csic.es; ramann@mpi-bremen.de; kostas.konstantinidis@gatech.edu

same species commonly show average nucleotide identity (ANI) of shared genes >95% between them and ANI <85% to members of other species[5-8]. Intermediate identity genotypes, for example, sharing 85–95% ANI, when present, are generally ecologically differentiated and scarcer in abundance, and thus should probably be considered distinct species[4,7,9] rather than representing cultivation or other sampling biases[10]. Sequence-discrete clusters similar to those described above for prokaryotes have recently been recognized for eukaryotic protozoa[11] and different types of viruses, including bacteriophages and viruses of eukaryotic hosts[12,13]. Therefore, it appears that similar species-level diversity patterns may characterize most microbes and viruses.

More recently, our team observed another discontinuity (or gap) in ANI values that may be used to define the units within a species, most notably genomovars and strains[14,15]. Specifically, the analysis of all complete isolate genomes ($n = 18,123$) from 330 diverse bacterial species revealed a clear bimodal distribution in the ANI values within the great majority (>95%) of these species. That is, there is a scarcity of genome pairs showing 99.2–99.8% ANI (midpoint at 99.5% ANI) in contrast to genome pairs showing ANI >99.8% or <99.2%[14]. We also suggested that the term genomovar could be used to refer to these 99.5%-ANI intraspecies units. We did not observe another pronounced ANI gap within the 99.5% ANI clusters, and thus recommended the use of the term strain only for nearly identical genomes based on the prevailing expectation that members of the same strain should be phenotypically very similar. Specifically, we proposed to define a strain as a collection of genomes sharing ANI >99.99% based on the high gene-content similarity observed among genomes at this high ANI level, e.g., typically, >99.0% gene content is shared (vs. ~90% gene content shared at 99.5% ANI)[15]. It follows that clones are organisms with identical (clonal) genomes, and thus a strain could encompass several clones[15]. These definitions are largely consistent with how several units within species such as sequence types and strains have been recognized previously, but provide units that encompass genomically more homogenous organisms compared to the existing practice and the means to standardize intraspecies definitions across taxa[14]. Accordingly, we use these definitions below for genomovars and strains. Furthermore, more recent work has revealed similar intraspecies diversity patterns for viruses of prokaryotic and eukaryotic hosts[16].

To better describe and model these diversity patterns, it is imperative to understand what mechanisms underlie the creation and maintenance of sequence-discrete species and genomovars; that is, how members of a sequence-discrete unit cohere together. Several competing hypotheses based on functional differentiation (*ecological species*), recombination frequency (*recombinogenic species*), or variations of these hypotheses, have been advanced to explain the 95% ANI species gap (or the newer 99.5% ANI gap within species) [reviewed in refs. 4,17,18]. Specifically, ecological speciation includes cases in which members of the same species (e.g., individual prokaryotic cells) could be functionally differentiated from members of other, related species (or genomovars of the same species) either due to specialization for different growth conditions or different affinities for the same energy substrate. Selection over time for these functions favors the growth of the corresponding members and may result in purging (loss) of diversity (e.g., elimination of members not carrying these functions) and thus speciation. Notably, given an estimated mutation rate of ~$4 \times 10^{-10}$ per nucleotide per generation[19] and between 100 to 300 generations per year[20], it would take two distinct lineages of a gut microbe such as *Escherichia coli* (*E. coli*) at least 100,000 years since their last common ancestor to accumulate 0.5% difference (i.e., fixed mutations) in their core genes or 99.5% ANI (in absence of any recombination). Therefore, given enough time, it is possible to have ecological purging of diversity even at around the 99.5% ANI level (let alone at 95% ANI) that accounts for the ANI patterns observed.

While a few examples of ecological speciation have recently been reported for prokaryotes based—primarily—on differential use of growth substrates[21,22], these were not directly associated with the prevailing sequence-discrete species recovered by the metagenomes. That is, these studies have shown that ecological speciation is possible but to what extent it accounts for the sequence-discrete units (or, in other words, is the prevailing mechanism in nature) remains to be evaluated. Further, these studies typically involved laboratory enrichment studies with strong selection pressures (e.g., high substrate concentrations), which may be rather different compared to natural conditions.

Members of a species (or a genomovar) could cohere together via means of unbiased (random) gene exchange which is more frequent within than between species. [Note that this is a fundamentally different mechanism than the sexual reproduction in eukaryotes and the accompanying biological species concept, although the ultimate outcome in terms of species cohesion may be similar. That is, gene exchange does not occur during a meiosis step but via vectors of horizontal gene transfer (HGT) mediated by recombination. Hence, we opted to not use biological or sexual species here, i.e., terms that are commonly used for Eukaryotes, and, instead, refer to them as recombinogenic species]. Indeed, several studies have concluded that the frequency of homologous recombination could be high enough to have a greater effect on sequence evolution than point (or diversifying) mutation, as has been shown to be the case for *Campylobacter* species[23] and other taxa more recently[24-26]. However, these studies have not been able to assess whether recombination is occurring across the genome (that is, it is not biased spatially and/or functionally) to serve as a force of cohesion or/and to what extent it accounts for the sequence-discrete units. In fact, at least in the *Campylobacter* case, recombination was convincingly shown to be biased to a few specific regions (genomic islands) of the genome and functions (e.g., antibiotic resistance and motility) that are apparently under strong positive selection, while there are several long segments of the genome that do not recombine. Thus, recombination is unlikely to lead to species cohesion in such cases, even if it appears to affect more sequence evolution overall than point mutations (e.g., recombination to mutation ratio >1), since the non-recombining segments of the genome will continue to diverge[27]. In summary, the question of whether or not recombination is frequent and random enough across the genome to maintain the sequence-discrete units identified in recent genomic and metagenomic surveys remains to be rigorously tested. By analyzing available closely related genomes here we show that recombination frequency coupled to ecological cohesiveness among members of these units might account for the species and intraspecies clusters observed previously.

## Results and discussion

### *Salinibacter ruber* genomes show rampant, genome-wide recombination when coupled to high ecological niche sharing

To obtain new insights into the role of recombination as a force of species/unit cohesion, we focused initially on a well-sampled bacterial species, *Salinibacter ruber* (*Sal. ruber*), which thrives in natural or engineered hypersaline environments, and subsequently evaluated the applicability of the resulting findings with a recently reported collection of *E. coli* isolate genomes originating within a ~100 km radius from livestock farms in the United Kingdom[28]. Engineered solar salterns are operated in repeated cycles of feeding with natural saltwater, increasing salt concentration due to water evaporation caused by natural sunlight, and finally, salt precipitation for human consumption. Previous studies have shown that salterns in different parts of the world harbor recurrent microbial communities each year[29,30]. These communities show low class/family diversity, generally consisting of two major lineages, i.e., the archaeal class Halobacteria class and the bacterial family of Salinibacteraceae, class Rhodothermia[30-32], but with

relatively high species richness within each class[33,34]. Notably, *Sal. ruber* makes up at least 1–2% of the total microbial community in salterns in any sample characterized to date, and typically between 5 and 25% of the total; that is, it represents a highly abundant member of the saltern communities and it is easy to isolate in pure culture[15]. To provide new insights into the functional role of intraspecies gene diversity, we have previously exposed the high-salt, high-sunlight adapted microbial communities at the end of the salt harvest cycle (~36% NaCl) in the "Es Trenc" solar salterns on the Island of Mallorca (Spain) to changing environmental conditions for about one month, and followed the communities with time-series shotgun metagenomics relative to control ponds with no treatment (i.e., ambient sunlight and salt-saturation conditions) during this period[34,35]. The changing conditions included an experimental manipulation of light intensity through the application of a shading mesh as well as (in separate ponds) lowering salinity to ~12% through the dilution of the brine with seawater. To aid the metagenomics, we isolated and sequenced 102 randomly selected isolates of *Sal. ruber* from the same samples. We supplemented this genome dataset with 20 isolates recovered from a single 1-liter sample—at salt-saturation point—from the Salinas del Carmen solar salterns on the Canary Islands (>2000 km away from Mallorca), 63 isolates from a single sample from Santa Pola's salterns (mainland Southern Spain, 300 km away from Mallorca) as well as nine available *Sal. ruber* genomes from the NCBI database (accession numbers and relevant metadata for all genomes used are provided in Supplementary Data 1). The ANI value patterns among all these genomes have revealed a pronounced gap around 99.5% ANI, consistent with the previous literature mentioned above and justifying *Sal. ruber* as a model system to study recombination patterns and speciation[15].

We compared the available *Sal. ruber* isolate genomes of varied genomic relatedness to each other, ranging from members of the same genomovar (ANI > 99.5%) to members of increasingly more divergent genomovars (ANI in the 97–99% range). The available genomes form six major clades (or phylogroups) based on ANI values or core-genome phylogeny, showing about 97.5–98% ANI among the phylogroups vs. >98% within a phylogroup (for members of different genomovars of a phylogroup), providing a gradient of relatedness (Fig. 1; phylogroups were defined based on the branching pattern of the core-genome phylogeny, and typically corresponded to >98% or >97.5% ANI within the phylogroup for *Sal. ruber* or *E. coli*, respectively). When we examined the nucleotide sequence identity patterns of individual genes across the whole genome, we observed that members of the same genomovar are typically identical or almost identical (nucleotide identity >99.8%) in most of their genes (>80% of the total, typically), except for a few regions (hotspots) that have accumulated substantial sequence diversity (e.g., showing 95–99% nucleotide identity to other members of the same genomovar; Fig. 2, top two genomes). Intriguingly, in about half of the cases, the genes in the hotspots of diversity have an identical match to another *Sal. ruber* genome of a different genomovar in our collection, indicating recent HGT events mediated by homologous recombination from that genomovar or its recent ancestors (Fig. 2, blue arrows; and Fig. S1 for examples of phylogenetic tree-based evaluation of HGT events). It is thus likely that the other half of the cases are also the product of recent HGT, but we did not have the donor genome among our isolate collection to confidently detect the HGT (i.e., find the >99.8% identity match). Consistent with this interpretation, our previous study indicated that at least a couple thousand genomovars make up the total natural *Sal. ruber* population in the salterns, most of which were low-abundance (rare) at the time of our sampling[15] and are not represented among the ~200 genomovars sequenced here (but could have served as donors in HGT in situ). Most of the genes in these hotspots represented core (shared) genes of the species although several accessory (or variable) genes were also noted. Alternatively, these divergent genes (and hotspots of diversity) could

represent regions of hyper-mutation, but this scenario appears less likely given that the predicted functions of the divergent genes are, more or less, random subsections of the total functions in the genome (Fig. 3) and do not show increased non-synonymous (pN) mutations (Fig. S2). Hence, the increased sequence diversity between members of the same genomovar is unlikely to represent hyper-mutation or positive (adaptive) selection. Further, the length of the (presumed) recombined segments, using the total length of consecutive recombined genes as a proxy, was similar to that observed in previous laboratory recombination studies[36] and ranged between 1 and 20 kbp, with the majority being 1–3 kbp long (Fig. S3). Therefore, it appears that these *Sal. ruber* genomes are engaging in genome-wide, rapid recombination that affects sequence identity much more than point mutations, revealing a recombinogenic rather than clonal sequence evolution.

A recombination event can increase similarity between the two partner genomes (by the removal of mutations), but it could also increase dissimilarity between the two when recombination occurs with a third genome that is more divergent [note that non-homologous recombination, mediated usually by mobile elements, also typically leads to increased dissimilarity, and is assessed below]. To further verify that most recombination events we noted above are cohesive (removal of mutations and increasing within-group similarity), as opposed to diversifying (introduction of new genetic material from outside the group that decreases within-group similarity), we classified genes into three groups for each pairwise genome comparison: recombinant genes (>99.8% identity within the genome pair considered, a proxy for cohesion force), recombinant genes with a 3rd partner (<99.8% identity within the genome pair considered but >99.8% identity with a 3rd genome, a proxy for diversification force) and non-recombinant genes (<99.8% identity with any genome in the dataset, proxy for point mutation). For each genome in our collection, we calculated the total number, length, and mismatches of genes classified in the three categories described above against every other available genome (Fig. S4). The main pattern observed (59% of total genomes evaluated) was recombination with a third genome belonging to the same phylogroup as the reference genome (phylogroup cohesive force). The second most common pattern was also dominated by recombinant genes with a third partner, but in this case, the partner genome was from a different phylogroup (26.5% of total, phylogroup diversifying force but cohesive at the species level). Non-recombinant genes involved only a minority of cases (14.5% of total), probably due to the under-sampling of the genome diversity of the species. These results suggested that recombination acts mainly as a cohesive force at least at the phylogroup level (for the genomovar level, see below).

To estimate more precisely the relative contribution of homologous recombination compared to point mutation, we developed an empirical approach based on the sequence identity patterns across the genome. Our approach identifies recombined genes that represent recent events, i.e., showing 99.8–100% nucleotide sequence identity, and subsequently calculates how much sequence divergence these events presumably removed based on the ANI of the genomes compared. For example, if the ANI of the two genomes compared is 97%, this would mean that the divergence of the recombined genes was about 3%, on average, before the recombination took place, and thus recombination should have removed (purged) a total of nucleotide differences that should roughly be equal to *3% × total length of recombined genes*. During the same evolutionary time, point mutation can create nucleotide differences that should roughly be equal to *average divergence of recombined genes × total genome length* (because we are only focusing on recent evolution that corresponds to the 99.8–100% identity threshold used to identify recent recombination events or 0.00–0.2% accumulated sequence divergence). [Note that >99.8% identity was used as the threshold because it corresponds to

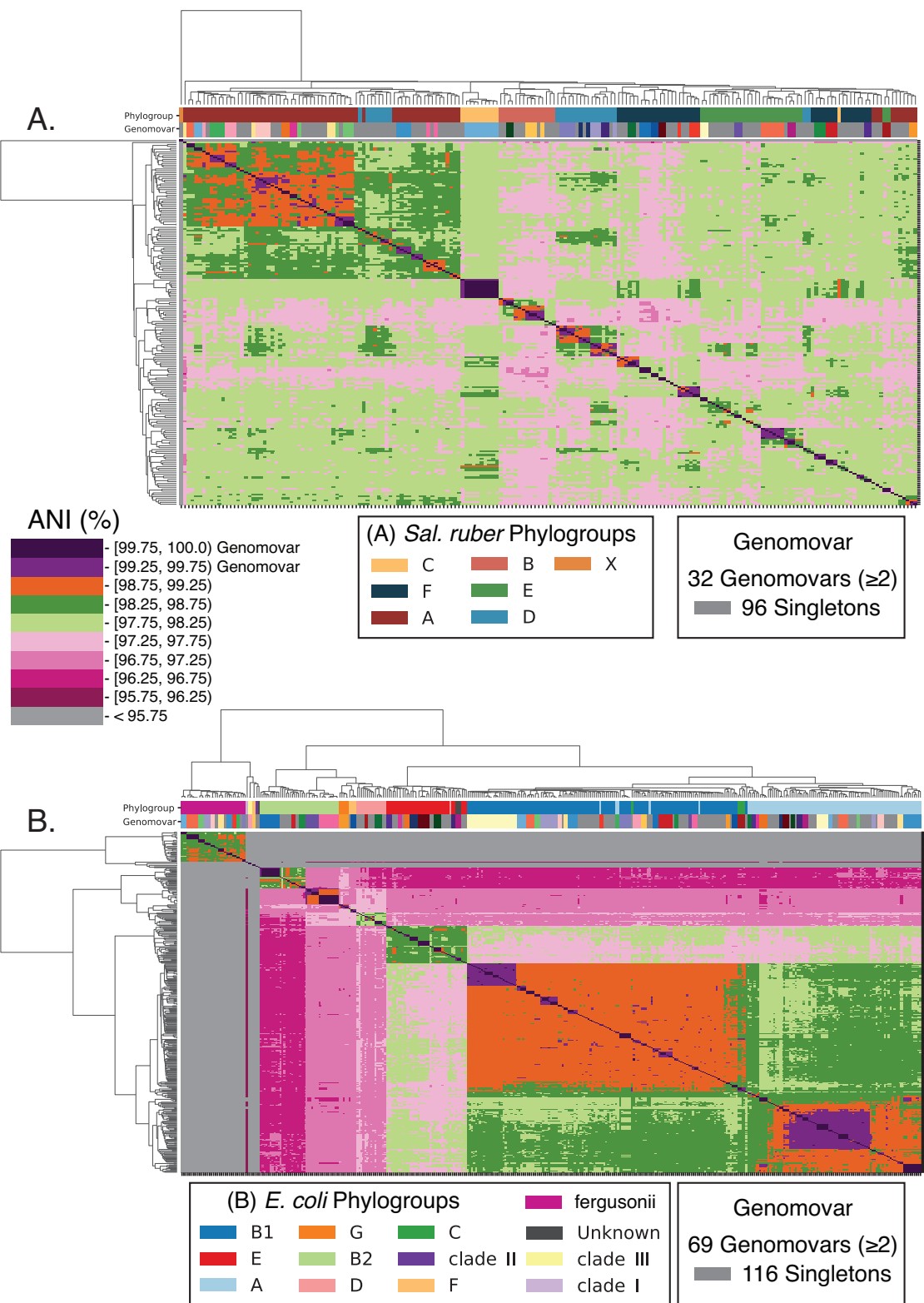

**Fig. 1 | ANI clustering showing genomovar and phylogroup structure for the *Sal. ruber* and *E. coli* genomes used in this study.** All vs. all ANI values were computed for *Sal. ruber* (**A**) and *E. coli* and relatives (*E. fergusonii/Escherichia* clades I–III) (**B**) using FastANI with default settings. Hierarchical clustering was performed with average linkage using Euclidean distances. Phylogroups were determined from a concatenated core gene tree for each species and with ClermonTyping (see figure key for details). Genomovar assignments were called based on ANI values (see figure key).

enough evolutionary time for measuring the effect of point mutation with our empirical approach and it provides enough signal over background identity for detecting recombination between genomovars; using 100% identity as the threshold did not change our conclusions substantially]. Using this approach, we observed that the ratio of mutations purged by homologous recombination ($r$) vs. mutations created by point mutation within the same time ($m$), or simply the $r/m$ ratio, to be higher than 1 and often around 3–5 for

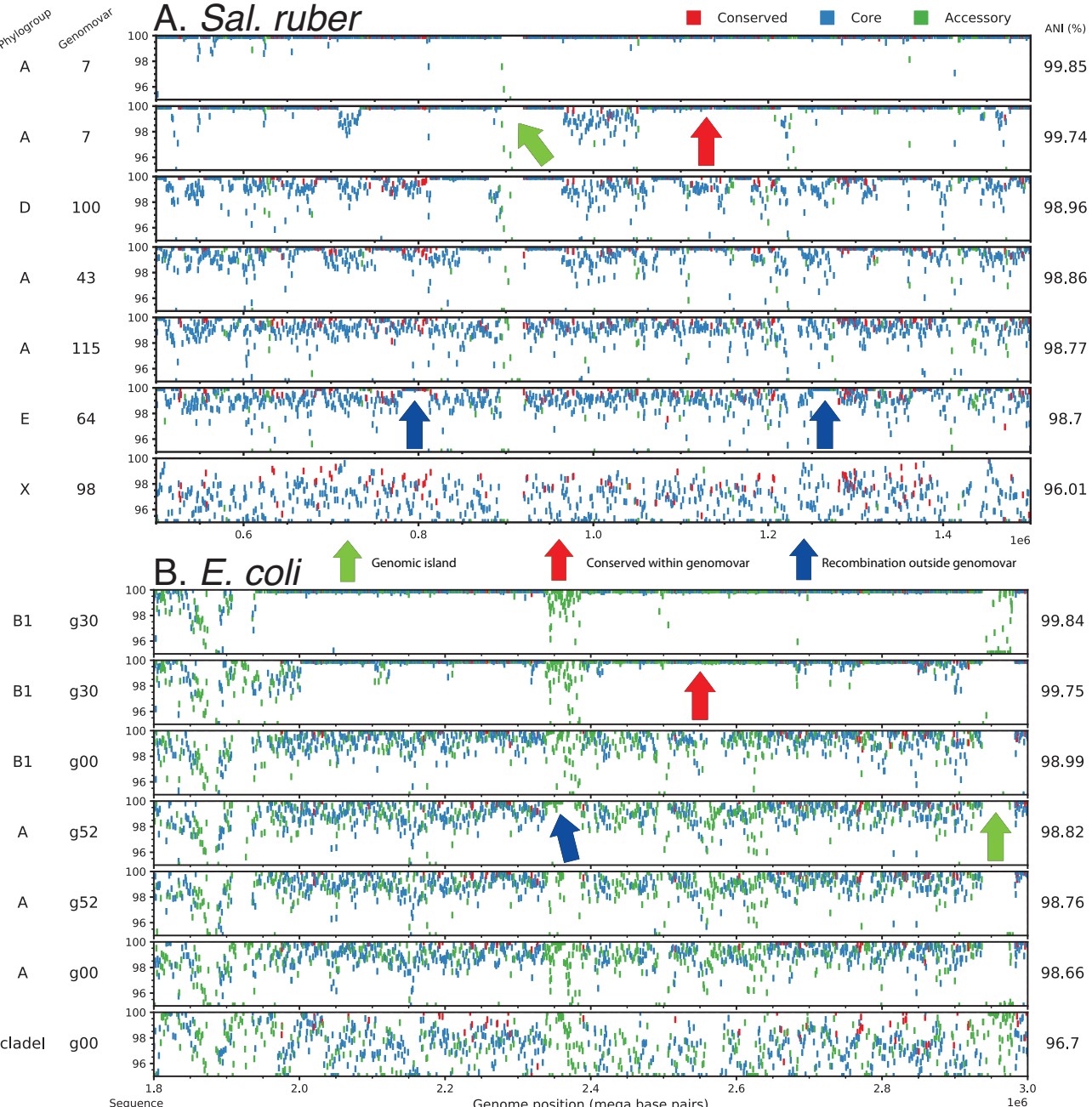

**Fig. 2 | Extensive recent recombination within the Sal. ruber and E. coli genomes.** Pairwise reciprocal best match (RBM) genes were identified for eight *Sal. ruber* (**A**) and eight *E. coli* (**B**) genomes spanning different genomovars and clades/phylogroups using BLAST+ with default settings. Each rectangular marker represents a gene, colored differently for highly conserved/universal, core, and accessory genes (see key), and represents the nucleotide sequence identity of RBM genes (*y*-axis) shared between seven query genomes (each row) and the same reference genome (*x*-axis, RBM gene position in reference genome) sorted by their ANI values to the reference genome shown on the far right of the panels. Two genomes from the same genomovar as the reference genome are shown in the top 2 rows and other genomovars and phylogroups are shown below. Note the hotspots of sequence diversity among members of the same genomovar, and that some of the genes in these hotspots show ~100% nucleotide identity between the reference genome and genomes of other genomovars (e.g., blue arrows). Green arrows denote genomic islands specific to the reference genome (i.e., not shared with query genomes, denoted by lack of markers in the genomes not carrying the island in the corresponding region of the reference genome) while red arrows denote highly identical regions conserved within the genomovar.

several genome pairs, especially members of different genomovars of the same phylogroup (Fig. 4; using the ANI of the non-recombined parts of the genome—as opposed to the whole genome—in the equation above provided even higher *r/m* ratios; data not shown for the latter). In contrast, the *r/m* ratio between *Sal. ruber* genomes and those of *Salinibacter pepae* (*Sal. pepae*), the closest known relative that often co-occurs with *Sal. ruber* in the salterns and shares high genetic (ANI~94%) and metabolic relatedness[37], was usually much lower than 1 (Fig. 4). Further, genomes of different phylogroups typically showed lower *r/m* values than genomes of the same phylogroup, often around 1 or even lower (denoted by points with ANI < 98% in Fig. 4). Note that it is not feasible to perform this type of analysis for members of the same genomovar due to the high identity across the whole genome (i.e., there is no signal over the background level of sequence identity to

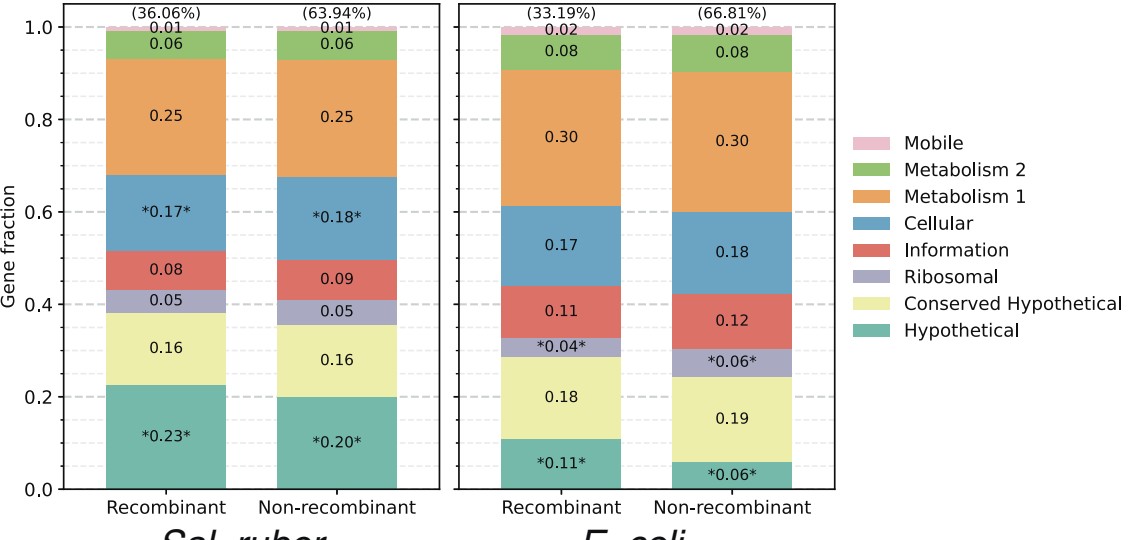

**Fig. 3 | Limited functional biases in the recently recombined genes.** The graphs show gene annotations summarized by high-level COG categories as a fraction of total genes in the genome (*y*-axis) for RBM genes divided into two categories (*x*-axis): genes with ≥99.8% sequence identity (recombinant), and genes with <99.8% sequence identity (non-recombinant). The asterisks represent functional categories found to be significantly different by one-sided Chi-square test (*p* value < 0.05) with Benjamini/Hochberg multiple test correction, likely reflecting genes undergoing more frequent recombination than the average gene in the genome, favored by selection for the corresponding functions. Nonetheless, note that, overall, all functional categories are subject to recombination (left columns) and, more or less, with the same frequency—or distribution—as they are found in the genome (right columns) for both species.

detect recombination), and thus our *r/m* ratio estimates for this level (i.e., ANI > 99.5%) are not reliable.

We also assessed the distribution of the genes across the genome identified as exchanged to reveal whether recombination affected all regions of the genome (random distribution), and could lead to recombinogenic species and unit coherence, or if instead the exchanged genes are spatially located in a few regions across the genome (biased distribution). The latter pattern would indicate selection-driven genetic exchange (not recombinogenic speciation), and ecological speciation. Our analysis showed that, for every region of the genome longer than 100–200 kbp, the importance of recombination was greater than point mutation in at least a couple pairs of genomes from different genomovars (Fig. S5), revealing that recent, rampant recombination has affected all regions of the genome. Further, while the fraction of the genome affected by recombination between any two genomes (of different genomovars) was almost always <50% of the total length (pairwise comparisons), when we compared one reference genome against representative genomes of all available genomovars, this fraction often approached 80% or higher when all recombination events detected with all possible partners in the analysis were summed (Fig. 5 and Fig. S6; one vs. many comparisons; note that an individual gene is counted only once for this analysis so as not to overestimate recombination regardless of how many genomes were found to have recently exchanged the gene). Such results were obtained with all reference genomes used in the analysis and did not appear to be specific to one or a few (reference) genomes or clades. That is, almost the whole genome was found to have recently recombined when all *Sal. ruber* genomes were considered in the analysis. Therefore, it appears that, for the *Sal. ruber* genomes evaluated here, homologous recombination is frequent enough and random (spatially across the genome) enough to serve as the mechanism for species cohesiveness. Further, we did not observe any strong biogeographical patterns (i.e., diversity to be locally constrained) in our recombination analysis, e.g., genomes from the Mallorca Island shared recent recombination events with genomes recovered from the Canary Islands or mainland Spain (Santa Pola) (Fig. S7). It appears that the latter result is attributable to the fact that the same genomovars can be found across these Spanish sites based on our preliminary analysis (Fig. S7; i.e., genomovars show little biogeography), although more rigorous testing of this hypothesis is needed because the number of available genomovars found across the three sites, and generally relative to the total *Sal. ruber* genomovars expected to be present in each site[15], is still rather limited. Therefore, it looks like that the recombination patterns reported here might be applicable to the global *Sal. ruber* population, not just the local population present in a single site (e.g., a saltern pond).

Our analysis also showed that genetic exchange between members of (distinct) genomovars of the same species is much more frequent than between members of different species, even after we accounted for the higher relatedness, and thus sequence identity of shared genes, among the former relative to the latter genomes on the derived results (e.g., Fig. 5, open circles). For instance, in the sampled salterns where *Sal. pepae* was indeed found to co-occur with *Sal. ruber* (ANI between the two species is ~94%)[37], our comparisons show that the two species rarely exchange shared (core) genes via a homologous recombination mechanism, at least ten times less frequent than within-species gene exchange (Fig. 5), consistent with the *r/m* ratio results mentioned above (Fig. 4) and the low efficiency of homologous recombination expected at this level of genetic relatedness. That is, recombination efficiency drops by about fivefold when the recombined sequences show ~99% nucleotide identity vs. 95% identity, and by tenfold with 90% identity[4,36]. Further, new-gene exchange that creates genomic islands through a non-homologous recombination mechanism mediated by mobile elements is much less frequent compared to shared-gene exchange via homologous recombination among the *Sal. ruber* genomes (Fig. 2). Therefore, shared-gene exchange via homologous recombination and similar, but not necessarily fully overlapping, ecological niches, appear to keep these genomes as members of the same, sequence-discrete *Sal. ruber* species, and accounts for the 95% ANI gap at the species level.

Since recombination between members of different genomovars is quite frequent based on our evaluation (Figs. 2, 5 and Fig. S7), we

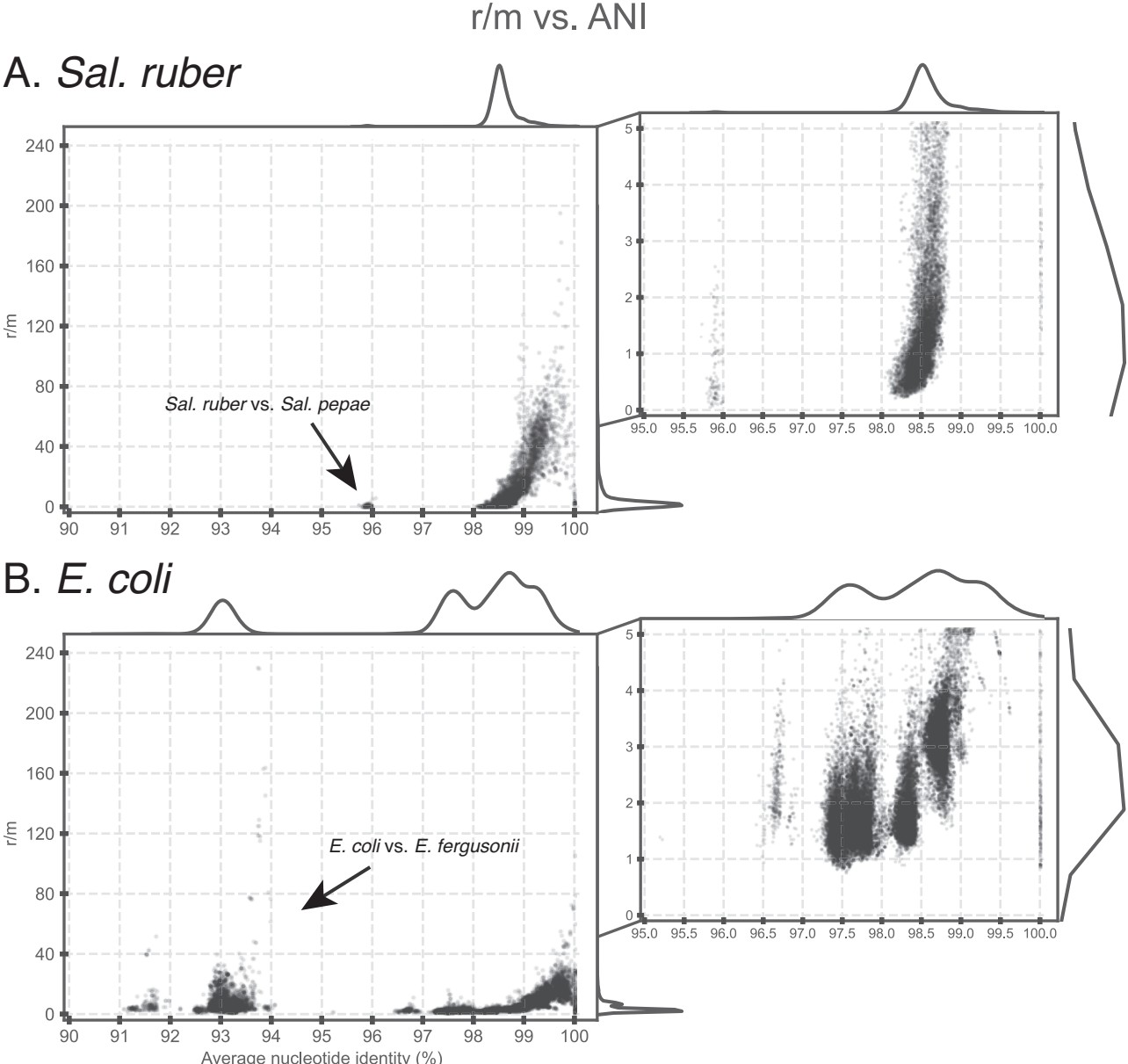

**Fig. 4 | Recombination to mutation (r/m) ratio as a function of the ANI of the genome pairs compared.** The *r/m* ratio (*y*-axes) was estimated for all genome pairs in our collection for each species (graph title on top) using the empirical approach described in the main text, and is plotted against the ANI value of the genome pair compared (*x*-axes). The marginal plots outside the two axes show histograms for the density of datapoints on each axis. Graphs on the right are zoomed-in versions of the main graphs on the left in the 0–5 range of the *y*-axis values. Top graphs (**A**) show results for *Sal. ruber* genomes; bottom graphs (**B**) show *E. coli* genomes. Note that the ratio is frequently above 1 for genomes sharing between 98.5 and 99.5% ANI (e.g., members of different genomovars of the same phylogroup) for both species and that the estimates above ~99.5% ANI are not reliable due to the inability to detect recombination at this high sequence identity level. A few outlier datapoints (genome pairs) with ratios higher than 100 were also observed in the 98–99.5% ANI range and are due to the high identity of the recombined genes identified (causing the denominator in the *r/m* ratio to be a small number); the graphs on the right show the majority of datapoints, and thus better represent the average pattern. Also, note that a few *E. coli* and *E. fergusoni* genome pairs (left part of the lower graph) show a ratio higher than 1, but this is driven by recombined genes that are localized in a couple of specific regions of the genome and encode specific functions (selection-driven recombination, and not widespread across the genome). See main text for additional details.

hypothesize that recombination is even more frequent within members of the same genomovar, and this may account for the high identity in the rest of the genome within a genomovar. This hypothesis is also supported by the fact that homologous recombination is known to be more efficient with higher sequence similarity[36], which is the case for members of the same (ANI > 99.5%) vs. different (ANI between 97 and 99%, typically) genomovars. Accordingly, our working model of how genomovars are maintained involves high recombination with members of the same genomovar when these members share the same ecological niche/habitat, and thus frequently encounter each other. And, this process accounts for (or leads to) the 99.5% ANI gap at the genomovar level. In contrast, when members of a genomovar are physically separated from each other, they may engage in recombination with co-occurring members of other genomovars (of the same species), which could then lead to the rapid emergence of new genomovars. Consistent with this model, we have evidence that at least some *Sal. ruber* genomovars show distinct ecological preferences; that is, several genomovars show abundances that correlate with low-salt

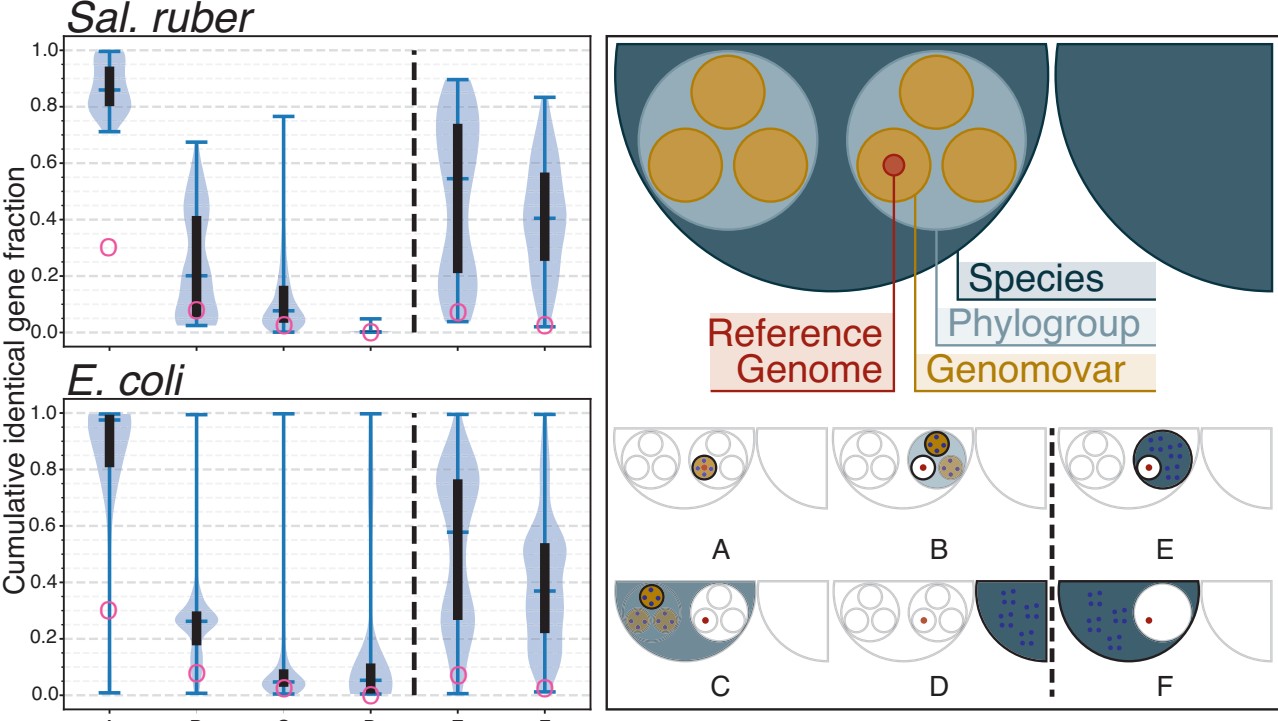

**Fig. 5 | Fraction of identical genes a genome shares with all other genomes within or between genomovar, phylogroup, and species.** Each genome was compared to all other genomes within each group (**A**–**F**) and the cumulative fraction of shared identical genes was recorded and plotted using the custom script *Allv_RBM_Violinplot.py*. The groups were as follows: **A** genomes within the same genomovar, **B** genomes in each separate genomovar within the same phylogroup, excluding genomes from the same genomovar, **C** genomes in each separate genomovar within different phylogroups, **D** genomes of the other species (*S. pepae* for *Sal. ruber* and *E. fergusonii* for *E. coli*), **E** genomes within the same phylogroup excluding genomes from the same genomovar, **F** genomes within the same species excluding genomes from the same phylogroup. Data are presented in hybrid violin plots where the top and bottom whiskers show the minimum and maximum values, the middle whisker shows the median value, the black boxes show the interquartile range, and the shaded light blue regions show the density of values along the *y*-axis.

The top graph shows results for *Sal. ruber* genomes; the bottom graph shows *E. coli* genomes. For *Sal. ruber* the number of genomes used in each group were, $n = 67$ for (**A**), 422 for (**B**), 897 for (**C**), 67 for (**D**), 176 for (**E**), and 192 for (**F**). For *E. coli*, $n = 199$ for (**A**), 2213 for (**B**), 2910 for (**C**), 425 for (**D**), 422 for (**E**), and 433 for (**F**). The right panel shows a graphical representation for comparisons performed for both graphs on the left. See also Fig. S6 for graphical examples of the underlying data. Note that while one or a few genomes create extreme outliers, overall, the fraction of identical genes gradually decreases among more divergent genomes compared. Also, note that our modeling analysis (red circles on the graph; see "Methods" section for more details) suggests—for example—that only about 6–7% of the total genes in the genome should be expected to be identical among genomes showing around 98.5% ANI if there is no recent recombination (i.e., the b and e groups); both species show many more such genes in one-to-one genomovar (group B) or one-to-many genomovars (group E) at this level, revealing extensive recent gene exchange.

concentrations and anti-correlate with concentrations close or at salt-saturation conditions while other genomovars show the opposite pattern [Ref. 15 and Fig. S8]. Therefore, it is conceivable that members of the same genomovar grow together when growth conditions for the genomovar are favorable, and consequently, there is more of a chance for recombination between them vs. with members of different genomovars that prefer different growth conditions during these periods, which eventually leads to the 99.5% ANI gap. Members of the same genomovar typically share higher gene content (0–10% of genes differ) relative to members of different genomovars (10–20% of genes differ)[14], which should account for higher ecological similarity. It should be mentioned, however, that we are currently unable to directly test this hypothesis (e.g., detect recombination events between members of a genomovar) because members of the same genomovar usually share ~100% sequence identity (no signal over the background identity level).

Alternatively, it is possible that the 99.5% ANI gap might be driven by the recent reproduction (a blooming event) of a few cells, members of the same genomovar, followed by rapid recombination of some of the offspring cells with members of other genomovars. Such inter-genomovar recombination events then lead to only a few genomes showing ANI values around 99.5% with the blooming (sub-)population

of cells since inter-genomovar recombination typically involves partners sharing <99% sequence identity, causing the quick divergence of the recombining genomes from the dominant sub-population. That is, recombination with other genomovars is the dominant process, not recombination within a genomovar, and has a genome/population-diversifying rather than a population-cohesion role under this hypothesis. We are currently unable to directly test this alternative hypothesis for the reasons mentioned above (e.g., low signal over the background identity within the genomovar level). However, it is intriguing to hypothesize that the same mechanism that drives the species and phylogroup gaps (where there is enough signal over background identity), with recombination being the force of cohesion as described above, may also drive the genomovar gap, as opposed to a distinct mechanism for the latter that involves diversifying recombination. Finally, while rapid and random (unbiased) diversifying recombination could theoretically provide sequence-discrete clusters similar to those observed for genomovars[38], obtaining clusters with the area of inter-cluster discreteness to be centered around the exact same ANI value (i.e., 99.5%) across many different taxa [Ref.14 and below] based on random processes seems unlikely. Hence, we favor the hypothesis that recombination as a cohesive force, coupled to high ecological cohesiveness, may be the mechanism that maintains not only the species

unit but also the intraspecies units revealed here and previously[14]. Consistent with this hypothesis, we commonly observed higher gene flow between genomovars of the same phylogroup (i.e., within the same phylogroup) than between phylogroups, although there are a few phylogroups with substantial inter-phylogroup gene flow as well (Fig. 5 and Fig. S7).

Finally, it is important to note that while the genomovars might have different growth preferences as we observed previously[15], these are likely not discrete but rather partially overlapping. For instance, we have isolated genomes that apparently prefer low salt from salt-saturation samples and vice-versa[15] and all these *Sal. ruber* genomovars can withstand salt-saturation conditions. Consequently, the 99.5% ANI gap might not always be clear or the gap may appear to be shifted to other ANI values in a few cases[14], and the gap is often not as pronounced as the 95% ANI gap that usually separates species (e.g., distinct species have less overlapping ecological niches than distinct genomovars or distinct phylogroups of the same species). Therefore, for future studies, we recommend assessing the ANI value distribution for the species of interest, and if the data indicate so, to adjust the ANI threshold to match the gap in the observed distribution. That is, we suggest performing all vs. all ANI computations and assess what ranges of ANI values correspond to peaks and valleys (gaps) in the resulting ANI distribution, which can subsequently be assigned to species, phylogroups, and genomovars. The gap values observed here for *Sal. ruber*, which are also highly similar to those for *E. coli* (below), should represent a reference point and will likely be applicable to additional, but not necessarily all, microbial groups.

## Applicability of the results to other species

To test how broadly these findings might apply to other bacterial species, we applied the bioinformatic framework outlined above to a set of available *E. coli* and *Escherichia fergusonii* (*E. fergusonii*) genomes isolated from livestock farms and runoff in the same region (~100 km radius)[28]. These *E. coli* genomes showed similar genomovar and phylogroup structure to the *Sal. ruber* genomes although there was a difference with three equally dominant phylogroups (B1, A, and E) among the former genomes vs. one dominant (and five less dominant) phylogroups among the latter genomes (Fig. 1). Patterns of gene exchange and *r/m* ratios for the *E. coli* genomes appeared remarkably similar to those of *Sal. ruber* based on the analysis of the identical gene fraction in one vs. many genome comparisons of genomovars of the same vs. different phylogroups (Figs. 2, 4, and 5). Notably, high levels of recombination among *E. coli* genomes, similar to those described above, have been recently reported by others based on independent approaches[25,26], but were not linked to the ANI-based units and/or shown to affect every segment of the genome as performed here. A few qualitative differences were also observed such as that *Sal. ruber* genomes showed extreme cases of high recent gene exchange between multiple phylogroups compared to *E. coli*, which showed only a few genome pairs with similarly high gene exchange and only between phylogroups B1, B2, and A (Fig. S7). There were also a few cases of high gene flow between *E. coli* and *E. fergusonii* genomes, the closest relative sharing about 93% ANI with *E. coli*, similar to the relatedness between *Sal. ruber* and *Sal. pepae*, but these appear to involve genes that are localized in a couple specific regions of the genome and encode specific (not random collections of) functions (selection-driven). These differences could be biological or ecologically meaningful; however, they could also be due to sampling bias (e.g., a different number of genomes is available for each phylogroup), and further research is required to investigate these differences.

In conclusion, our results show that recent gene exchange is both frequent and random enough across the genome to serve as the mechanism of cohesion for the sequence-discrete units of at least the two taxa studied here, *Salinibacter* and *Escherichia*. Recent gene exchange appears to be mediated by homologous recombination at the genetic/molecular level and by high ecological similarity for bringing the organisms in close, physical proximity for the genetic exchange to take place. While elements of this model have been proposed previously [e.g., refs. 17,25,39], it is important to note that we provide a complete mechanistic view of how the evolution of species, phylogroups and genomovar units takes place and the necessary quantitative data in support of the model. Specifically, recombination was previously hypothesized to serve as the mechanism of species cohesion[18], and be associated with the ecological differentiation of subpopulations[17], but unambiguous data in support of this hypothesis has been elusive. Indeed, several previous studies show that recombination may contribute more to sequence evolution than point mutation[25,40,41] but these studies were not able to assess whether recombination was spatially and functionally unbiased across the genome, did not link recombination to the ANI units, or did not consider the role of ecological relatedness at the genomovar level as the latter was assessed here. The methodology developed here effectively circumvents these limitations. Further, instead of attributing the sequence-discrete units to either ecological or genetic (e.g., recombination) mechanisms as the prevailing theories of microbial speciation do[18,42], our model suggests that the two types of mechanisms may operate together, which represents a departure from previous models of speciation. Our results also suggest that bacteria, and likely other microbes, may evolve more "sexually" than previously thought. The drastically different lifestyles of the two taxa studied here, with *E. coli* being a human/animal gut commensal and *Sal. ruber* a halophilic environmental bacterium, as well as their large phylogenetic distance, as members of distinct bacterial phyla, indicate that the results reported here are likely applicable to additional taxa. In fact, our recent work suggests that similar patterns of recent gene exchange can be observed in human and bacterial viruses[16]. Therefore, it is highly likely that the model of genome evolution and speciation proposed here applies more broadly in the microbial world.

The modes of HGT, e.g., transformation, transduction or conjugation, and especially the relative importance for the HGT events detected, in the *Sal. ruber* or the *E. coli* cases remain unknown at present and should be the subject of future research toward a more complete understanding of the evolution of discrete units. Whether the high frequency of homologous recombination we observed between these genomes gives a selective advantage over evolutionary time or instead is just a side effect of processes that are independent of the creation of discrete units such as DNA repair of errors/mutations from UV or other damaging agents, or DNA replication also remains to be determined. Regardless of what the underlying reason(s) for the high frequency of recombination are, or which modes of HGT are dominant, our results clearly show that recombination, coupled to niche overlap, underlies the creation and maintenance of discrete units.

## Methods

All genomes were downloaded from NCBI's Assembly database. Accession numbers are available in Supplementary Data 1. The *Sal. ruber* genomes reported here that were isolated from Santa Pola were sequenced as part of the GEBA V project. Step-by-step details for our main analysis workflow are outlined in a GitHub repository: https://github.com/rotheconrad/F100_Prok_Recombination (and https://doi.org/10.5281/zenodo.13922077). Briefly, ANI was calculated with FastANI v1.33 with default settings[43]. Phylogroups for *Sal. ruber* were determined from a concatenated core gene tree and hierarchical ANI tree. Phylogroups for *E. coli* were retrieved from Shaw 2021 who assigned them with ClermonTyping v1.4.1[44]. Genomovar assignments were called manually based on hierarchical clustering of ANI values. Hierarchical clustering was performed in Python 3.6+ using Seaborn v0.12.1[45] and SciPy v1.9.3[46] function scipy.cluster.hierarchy.linkage with parameters method = "average", metric = "euclidean". Reciprocal

best-match genes were computed using BLAST+ v2.13.0 with default settings[47]. Gene predictions were called using Prodigal v2.6.3 with default settings[48]. Gene clustering was performed with the cluster module of MMSeqs2 with settings --min-seq-id 0.90 --cov-mode 1 -c 0.5 --cluster-mode 2 --cluster-reassign[49]. Recombined genes and the ratio of recombination-to-mutation were determined as described in the main text.

Our modeling analysis to estimate the fraction of identical genes expected between two genomes of a given ANI value without any recent recombination between the genomes (i.e., red circles in Fig. 5) used the following approach. A random ancestral genome sequence was generated with 3000 genes of variable length (selected from a normal distribution; mu = 1000 bp, stdev = 250) with 10 bp-long random sequence inserted between any two genes. Subsequently, daughter genomes were generated by the addition of random, single-nucleotide mutations to ancestral genes to match a gamma distribution for RBM gene sequence identity with the distribution mean fit to the desired ANI of the genome pair. We generated ten daughter genomes from the ancestral genome for each ANI value in the range of 95–100% ANI with a step size of 0.01 for a total simulated population size of 5001 genomes. The resulting frequency values matched well the average frequency of such identical genes found between randomly drawn genomes from NCBI of similar ANI, which are expected to not show extensive recombination, indicating that our population genome simulation was robust. The code used to simulate these genomes is available at: https://github.com/rotheconrad/Population-Genome-Simulator (and https://doi.org/10.5281/zenodo.13922083).

### Methods of Supplementary figures

**Tanglegrams (Fig. S1).** Regions of between-phylogroup recombination were identified using the *Sal. ruber* graph in Fig. 2A. We selected core genes from these regions (~800,000 bp and ~1,250,000 bp) and extracted the gene sequences from the Prodigal files using seqTK v1.3-r117 (available at: https://github.com/lh3/seqtk.git). Sequences were aligned with MUSCLE v3.8.31[50] and evaluated to ensure they were of good quality. MEGA11[51] was used to generate individual maximum likelihood trees that were compared to the *Sal. ruber* ANI cladogram shown in Fig. 1 and Fig. S1. Tanglegrams were drawn in R using the ape v5.7.1[52], dendextend v1.17.1[53], and phylogram v2.1.0[54] packages.

**pN vs. pS (Fig. S2).** The reference genome for Fig. 2A was used to calculate the difference between synonymous and non-synonymous substitutions to measure the effect of selection on the genes. The calculations involved 13 genes spread across the genome at intervals of ~250,000 bp and 20 genes from the same regions at intervals of 50,000 bp. Gene sequences were extracted from Prodigal files using seqTK and aligned by codons using the Clustal[55] module built into MEGA11. The Kumar model[56] was used to calculate overall mean distances for the 13 genes spread across the genome and pairwise distances for the genes in the region of interest as implemented in the script developed by T. Zhu, available at https://github.com/zhutao1009/dnds.git. The plot was generated in Python.

**ANI trees (Fig. S7).** All-vs-All ANI tests were done with FastANI[43] for *Sal. ruber* and *E. coli* independently. These data were used to construct ANI-based cladograms in R using the Euclidean distance and average cluster methods. Ape v5.7.1[52] was used to convert the cladograms into Newick-formatted trees that could be uploaded to iTol[57] for further annotation. Phylogroups were annotated according to the major monophyletic clades that could be identified in the ANI tree (for *Sal. ruber*) and according to the groups previously identified in literature (for *E. coli*). F100 scores generated by the pipeline were used to draw connection arcs between pairs of genomes as a representation of the frequency of recombination between two genomes.

### Reporting summary

Further information on research design is available in the Nature Portfolio Reporting Summary linked to this article.

## Data availability

Accession codes for the genomic sequence datasets analyzed in this study are provided in Supplementary Data 1. Other data are available in the main text or the Supplementary Information document.

## Code availability

Our main analysis workflow is available at: https://github.com/rotheconrad/F100_Prok_Recombination. (https://doi.org/10.5281/zenodo.13922077) and https://github.com/rotheconrad/Population-Genome-Simulator (https://doi.org/10.5281/zenodo.13922083); and for Supplementary analysis/figures, https://github.com/catbrink/Explaining-ANI-gaps-Code-for-supplementary-figures.git.

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

## Acknowledgements

The work (proposal https://doi.org/10.46936/10.25585/60001079) conducted by the U.S. Department of Energy Joint Genome Institute (https://ror.org/04xm1d337), a DOE Office of Science User Facility, is supported by the Office of Science of the U.S. Department of Energy operated under Contract No. DE-AC02-05CH11231 and the US National Science Foundation (Award No 1831582 and 2129823) to KTK. The authors would like to thank the whole team at Salinas d'Es Trenc and Gusto Mundial Balearides, S.L. (Flor de Sal d´Es Trenc) and of the Salinas

de Fuerteventura (Salinas del Carmen) for allowing access to their facilities and their support in performing the experiments. The research at the IMEDEA was funded by the Ministry of Science and Innovation projects PGC2018-096956-B-C41 and PID2021-126114NB-C42, both also supported—in part—by European Regional Development Fund (FEDER) funds, and through the "Maria de Maeztu Centre of Excellence" accreditation to IMEDEA (CSIC-UIB) (CEX2021-001198). KTK's research was supported, in part, by the U.S. National Science Foundation (Award No. 1831582 and No. 2129823). RRM acknowledges the financial support of the sabbatical stay at Georgia Tech supported by the grant PRX18/00048 of the Ministry of Science and Innovation. TV acknowledges the "Margarita Salas" postdoctoral grant, funded by the Spanish Ministry of Universities, within the framework of the Recovery, Transformation and Resilience Plan, and funded by the European Union (NextGenerationEU), with the participation of the University of Balearic Islands (UIB). T.V. and R.A. acknowledge support by the Max Planck Society.

## Author contributions

Conceptualization: K.T.K., R.A., R.R.M., L.M.R., and R.E.C. Data curation: R.E.C., C.E.B., T.V., and S.N.V. Formal analysis: R.E.C. and C.E.B. Visualization: R.E.C. and C.E.B. Funding acquisition: K.T.K., R.R.M., and R.A. Investigation: R.E.C., C.E.B., L.M.R., T.V., and B.A.R. Methodology: R.E.C., C.E.B., L.M.R., T.V., and B.A.R. Project administration: K.T.K., R.R.M., and R.A. Supervision: K.T.K., R.R.M., R.A., and S.N.V. Writing—original draft: K.T.K., R.E.C., C.E.B., and B.A.R. Writing—review and editing: R.E.C., C.E.B., T.V., L.M.R., B.A.R., J.K.H., S.N.V., R.A., R.R.M., and K.T.K.

## Funding

## Competing interests

The authors declare no competing interests.
