## [Transparent Peer Review file · Nature Communications]

Microbial species and intraspecies units exist and are maintained by ecological cohesiveness coupled to high homologous recombination

Corresponding Author: Professor Konstantinos Konstantinidis

Version 0:

Reviewer comments:

Reviewer #1

(Remarks to the Author)

Dear authors,

I read your MS with great pleasure in interest. Upon reading my comments below, it should be noted that I am not an expert on any microbial species in particular (esp not *Sal. ruber*). As a modeller, I consider my bioinformatic skills to be above the basic level, but not more so. Please keep that in mind when reading my comments.

In this review, you address an important problem: given the frequent exchange that is being shown by many recent papers (including your own), what happens to the microbial species concept, and which lines can be drawn? You discuss this in light of two opposing pressures: diversifying (point) mutations and cohesifying (is that a word?) recombination. You show that there are very strong cohesifying effects. If I understood correctly, you are even inferring fairly recent HGT events between isolates from geographically distant sites, but that this mostly happens within genomovars/phylotypes. This would imply your species (or microbes in general) are a) frequently exchanged between habitats, and b) that within this habitat they only recombine with their own genomovar/phylotype. The latter could be due to having specific niches (a proximity effect), or more farfetched reasons such as biased DNA uptake (seems to need a lot more assumptions). I find this concept a remarkable and novel insight, which although previously expected was never shown clearly. Although as you note you are also limited in scope by your data, you have set the stage for exciting new studies.

Here are a few points where I got stuck:

> I perhaps do not agree entirely on your focus on identifying "the real mechanism" behind these diversity patterns, as I'd rather see a good description of the data than read another discussion on what a species even is. But that is a philosophical discussion that is not really necessary here.

> You refer to HGT being mostly driving by HR on line 118. I agree that this is a likely mechanism, but some MGEs seem to move in ways that may look like HR. E.g. some TEs can excise and leave behind their original target site, such that a new TE will quickly take its place. I'm not sure it is necessary to discuss this in detail, but perhaps a brief mention on the possibility that this is driven by MGEs is good

> In line 122-124 you write "a previous study indicated". As I didn't take the time to actually follow up on the reference (most readers won't, so let me play an actual reader of your paper), please elaborate what you mean here. Is it that you infer X amount of genomovars by extrapolating from the current rate at which you could infer donors? I am confused.

> Line 139: I really like this distinction between recombination resulting in cohesiveness vs. divergence, and I think it deserves a better priming in the introduction (its present already, but its limited). Do you think your observation of HGT being mostly a cohesive force is related to HGT acting as a repair for Muller's Ratchet / gene maintenance mechanism?

> I would be careful in unpacking what is implied by local populations being relatively un-influential when it comes to recombination patterns. Some people use local to mean "both strains were isolated from the Black Sea", whereas others mean it came from the same dental cavity. Since the former describes millions of niches, and the latter describes a few at most, this difference matters for your interpretation. I think this part of the text (240-254) addresses this only partially, but should be carefully addressed.

> Why is Figure S5 not part of the main text? If you are limited by figures, could you make a simplified version of Figures S5 be part of the main text? This also relates to my last point

> I think the last sentence of the *Sal. ruber* section deserved much more emphasis throughout the entire MS: "we recommend using ANI value distributions". While you are probably not the first to say this, you are one of the first to highlight how important it is in this particular metric. In the age of "big data", I am perplexed at how many single-value things are being measured to this day, even when we know how misleading they can be. So my general advice would be to let this message shine a bit more in the abstract/introduction, although that is of course ultimately your own choice.

Minor points:

> Please change the colours for Figure 1 and 2 to be colour-blind friendly.

> Figure 4 is a bit crowded. Perhaps add some bigger margins because there is a lot going on.

Reviewer #2

(Remarks to the Author)

I agree with the authors that it is imperative to understand what mechanisms underlie the creation and maintenance of sequence-discrete species and genomovars. However, I found the exact hypothesis and methods of this study poorly explained in the (long) introduction. The discussion on clones, strains, genomovars, species is quite involved and could benefit from a Figure depicting the ANI cut-offs between these classes proposed by the authors. The authors state that both ecological differentiation and recombination influence divergence between sequence clusters which is very true, but I do not recognise a clear research question that explains how to distinguish between these forces. What does the sentence "recombination frequency coupled to ecological cohesiveness among members of these units might account for the species and intra-species clusters observed previously." actually mean? I also disagree with some of the assumptions in the intro, see comments below.

A lot of the text in the Results seems Introduction to me. Also, the lack of any subheaders makes it more difficult to grasp the main thrust of the argument. There does not seem to be a lot by way of data when reading the text (e.g. "Therefore, it appears that these *Sal. ruber* genomes are engaging in genome-wide, rapid recombination" does not seem to be supported by any statistics/formal analysis). The Figures show some interesting data but since many researchers have done comparative genome analyses (esp on *E. coli*), what is the added insight presented here exactly? For instance, the fact that genetic exchange is frequent, and also that it is more common between more closely related isolates is convincingly shown here, but this is not exactly a novel result. The high frequency of recombination indicates that it is a force for cohesiveness, but any evidence (dis)proving 'ecological selection' as a force of cohesiveness/divergence (requiring detailed, hard to collect data from the field) is much weaker. The authors state "instead of attributing the sequence-discrete units to either ecological or genetic (e.g., recombination) mechanisms, our model suggests that the two types of mechanisms may operate together, which represents a departure from previous models of speciation" but I just do not see this supported by the data. It is not explained why the focus is on the two chosen species. Demonstrating a general pattern requires a range of species but an in-depth species-level analysis is also useful, however, inclusion of a second species is almost a distraction. The results introduce the concept 'phylogroup' which is yet another subdivision and one that was not even mentioned in the introduction, which I found confusing.

L4

"Recent large-scale surveys of prokaryotic communities (metagenomes) as well as isolate genomes have revealed that their diversity is predominantly organized in sequence-discrete clusters or units that may be equated to species. Specifically, genomes of the same species commonly show average nucleotide identity (ANI) of shared genes >95% between them and ANI <85% to members of other species. Intermediate identity genotypes, for example, sharing 85–95% ANI, when present, are generally ecologically differentiated and scarcer".

Is his not circular reasoning? Sequence clusters here are defined according to species status, rather than the other way around. Also, what is 'ecologically differentiated' based on? (gene content? Presence/absence across samples? Fitness measurements across different conditions?)

L43

"Notably, given an estimated mutation rate of $\sim 4 \times 10^{-10}$ per nucleotide per generation and between 100 to 300 generations per year [20, 44], it would take two distinct lineages of a gut microbe such as *Escherichia coli* at least 100,000 years since their last common ancestor to accumulate 0.5% difference (i.e., fixed mutations) in their core genes or 99.5% ANI"

But SNPs used in ANI calculations are also introduced via homologous recombination, which is around 3.6 times as frequent (Torrance *et al* 2024 PNAS) so this calculation is not correct.

L46

"Therefore, given enough time, it is possible to have ecological purging of diversity even at around the 99.5% ANI level (let alone at 95% ANI) that accounts for the ANI patterns observed."

I assume 'ecological purging' here equates to purifying selection. Why would all SNPs be purged though? Not all will be very deleterious.

L57

"[Note that this is a fundamentally different mechanism than the sexual reproduction in eukaryotes and the accompanying biological species concept, although the ultimate outcome in terms of species cohesion may be similar. That is, gene exchange does not occur during a meiosis step but via vectors of horizontal gene transfer mediated by recombination. Hence, we opted to not use biological or sexual species here and, instead, refer to them as recombinogenic species]."

Of course meiosis is very different from transduction/transformation/conjugation but genetic exchange in both eukaryotes

and prokaryotes DOES act as a cohesive force, as the authors note themselves, so I do not understand this sidenote. (I also note that the later "Our results also suggest that bacteria, and likely other microbes, may evolve more "sexually" than previous thought." seems to obviate this view.)

L62

"indeed, several studies have concluded that the frequency of homologous recombination could be high enough to have a greater effect on sequence evolution than point (or diversifying) mutation, as has been shown to be the case for *Campylobacter* species and other taxa more recently."

There are both older (more original) and newer (better) studies available that clearly demonstrate this point (which the authors do not take into account in their divergence time calculation above). Indeed, it is the norm, as Torrance et al show "For most species (82%), the homologous recombination rate was found to be more impactful on genomic evolution than mutations alone ($r/m > 1$)."

L65

"However, these studies have not been able to assess whether recombination is occurring across the genome (that is, it is not biased spatially and/or functionally) to serve as a force of cohesion or/and to what extent it accounts for the sequence-discrete units."

The Torrance et al study is based on WGS data so this statement is not correct.

L67

Regarding the general point "recombination is unlikely to lead to species cohesion in such cases since the non-recombining segments of the genome will continue to diverge" and genomic islands: GIs rapidly change in gene content and the point about ANI divergence does not apply here, the core genome will be reflective strain/species divergence, which at least in *C. jejuni* is affected by a high rate of HR (see again the Torrance reference).

L167

"mutations created by point mutation within the same time (m)," does this method take into account SNPs introduced by HR?

L291

biased functions?

L310

The authors cannot just state "In fact, our recent work suggests that similar patterns of recent gene exchange can be observed in human and bacterial viruses" without any explanation/evidence.

Abstract

I think the Abstract is very unclearly phrased. It ends with the authors stating that they have found the answer for what a species is, but what this answer is, remains very hazy. Also, 'persistent': what does this mean in this context? Detected in chronological sequencing studies? 'a novel methodology to identify recent recombination events': this needs a little bit of detail, what is novel about it? (this point is not explained in the Introduction either) 'unbiased (i.e., not selection driven) horizontal gene flow': I take issue with this, as HGT itself is followed by selection rather than is driven by it (unless the authors mean that there is second-order selection for HGT, which is not obvious here). 'diversifying mutation' is not a term in general use so please just use 'point mutation'

Reviewer #3

(Remarks to the Author)

Conrad, Brink et al, present a model that ecological cohesiveness coupled with high rates of recombination are responsible for sequence discrete units. Their work largely focuses on the analysis of their model organism, *Salinibacter ruber*, a common community member found in salterns. They apply their findings to a genomic collection of *Escherichia coli* isolates to ascertain if their observations are generally applicable to other bacteria. More specifically, they analyse their genomic collections to observe the role of unbiased homologous recombination and its frequency. In the latter half of the results and discussion they hypothesize what their observations could mean to their current working model of the emergence and maintenance of sequence discrete units. The discussion of what is a species in the context of bacteria or archaea is always intriguing and not, unsurprisingly, without a deal of hypothesizing as is the case in this manuscript. Some clarity or modifications to the methods and expansion of the discussion would improve the manuscript. Below are some comments which I hope the authors will find useful.

In this study the authors discuss recombination in genomovars and phylogroups. The definition of a phylogroup for *Salinibacter ruber* is given by a single line in the methods (line 320-321). How the phylogroups were defined should be more explicitly stated. Moreover, as figure 1 shows the phylogroups for both *Sal. ruber* and *E. coli* are not of equal size and diversity. Lines 137-153 discusses recombinant genes within and between different phylogroups. Here the size of the phylogroup should be taken into consideration as a larger phylogroup would mean more pairwise intra-phylogroup and less to inter-phylogroups comparisons compared to less populated phylogroups.

In the definition of genomovars the authors refer to this as 99.5%-ANI (line 21) I know this is the midpoint, however at line 108 genomvar is given as ANI >99.5% and yet more values are given in figure 1 for genomvars. It is a small point but being more consistent would greatly aid the reader.

The cutoff of using 99.8% ANI as the recombination threshold needs more justification. First, the authors have not provided a rationale for selecting 99.8% as the recombination cutoff (line 141). It appears this threshold might have been chosen because 99.8% represents the upper limit for defining a genomovar. However, the manuscript neither mentions this assumption nor justifies it. Second, the authors do not present an underlying null model in most of the recombination rate/fraction estimations. While using identical regions as indicators of recombination is common, this approach is typically applied to genomes from different species, where the genetic distance is sufficiently large, making the likelihood of identical regions arising from vertical inheritance very small (references 1, 2 below). However, since the authors are comparing genomes within the same phylogroup versus those from different phylogroups—and considering that genomes within the same phylogroup can exhibit ANI values greater than 98.5%—it's crucial to consider how genes with ANI greater than 99.8% could be shared between two genomes by vertical inheritance alone. For instance, a 1000 bp gene at 99% ANI has a 0.2% chance of being >99.8% ANI due to vertical inheritance, not accounting for the fact that some genes are more conserved due to negative selection. A few places where comparing against a null model that includes only vertical inheritance (and ideally considers negative selection) could improve the analysis: A, Comparing the r/m ratio of genomes within a phylogroup versus between phylogroups (lines 172-173, 209-210): Here, it is essential to include a null model because genomes within each phylogroup have higher ANI than those in different phylogroups. Thus, even with complete vertical inheritance, a larger genome fraction is expected to exceed 99.8% ANI, which could be mistakenly identified as recombination. B, calculating overall recombination (sum of 1 vs. all possible partners) for every genome (Lines 189-190): In this case, the estimated amount of recombination between genome pairs is summed across ~200 genomes. Therefore, even if vertical inheritance only leads to 0.1% of the genome being >99.8% ANI, this could falsely identify 20% of the genome as recombined.

The authors emphasize (i.e. abstract, conclusion section) that their model suggests that ecological or genetic mechanisms operate together to form sequence discrete units, and the model is a departure from previous models that emphasize one or another. This might be a little unfair to others, since the previous models cited by the authors clearly state that recombination alone is not sufficient for forming stable clusters that could lead to speciation (reference 3 below), or that the gradual decline of recombination with divergence is an important step in completing the ecological speciation process (reference 4 below), it fact there appears the author's model has similarity to what is proposed ecological speciation in (reference 4 below). Therefore, perhaps an expansion of the discussion to include previous works would be warranted.

1. Smillie CS, Smith MB, Friedman J, Cordero OX, David LA, Alm EJ. Ecology drives a global network of gene exchange connecting the human microbiome. *Nature* 2011; 480: 241–244.
2. Sheinman M, Arkhipova K, Arndt PF, Dutilh BE, Hermsen R, Massip F. Identical sequences found in distant genomes reveal frequent horizontal transfer across the bacterial domain. *eLife* 2021; 10: e62719.
3. Fraser C, Hanage WP, Spratt BG. Recombination and the Nature of Bacterial Speciation. *Science* 2007; 315: 476–480.
4. Shapiro BJ, Polz MF. Microbial Speciation. *Cold Spring Harb Perspect Biol* 2015; 7: a018143.

Version 1:

Reviewer comments:

Reviewer #1

(Remarks to the Author)

I am happy with the current state of the manuscript.

Reviewer #2

(Remarks to the Author)

I was surprised to see the positive assessments of the other two reviewers (and editor) after the first round of review.

Just looking at the revised Abstract (after all the section of the paper highlighting why the paper is novel and worthwhile):

“Ecological cohesiveness was inferred based on higher similarity in temporal abundance patterns of genomes of the same vs. different units, while recombination frequency was shown to be genome-wide and to have two times or greater impact on sequence evolution than point (diversifying) mutation.”

If the authors indeed would demonstrate a rise or fall in frequency of related sequence types over time, this would indeed have been a very interesting demonstration of ecological cohesiveness. However, none of the figures show this result! (Indeed, the term ‘temporal’ is not even mentioned anywhere else in the paper – it seems that is based on anecdotal evidence only – not sure where in the long paper).

That recombination is a force of cohesiveness is well-known and its impact has been shown to be even greater in other taxa. Contrary to what the authors assert, other researchers have taken a genome-wide approach to detecting homologous recombination rate, e.g. see <https://doi.org/10.1093/gbe/evs043> (published 12 years ago!). (‘recombination frequency was genome-wide’ is not correctly phrased if meant to mean that recombination frequency was similar across the genome btw.)

Why the authors feel that they can continue on with the statement “therefore, our results represent a departure compared to previous models of microbial speciation that invoke either ecology or selection-driven recombination, but not their synergistic effect.” is completely unclear to me. Many have already argued that ecological cohesiveness and recombinational

cohesiveness go hand in hand: for instance, by occupying the same spatial niche, there is a higher chance of cell-cell (or cell-eDNA) contact between different intra-species clusters. (Also, 'diversifying mutation' is not generally used as a term (or used at all) and is confusing as any mutation results in new diversity (homoplasies aside).)

This is just the Abstract. I have no desire to revisit all the other arguments that the authors have not bothered to properly address.

Reviewer #3

(Remarks to the Author)

Thanks to the authors for the revised manuscript and response to the queries and comments. There are two points to the authors. One which hopefully adds clarity to the original review comments and the other just a textual recognition of previous work.

Apologies for not being clear. The null model, sorry about missing out on the null model part. However, I still have a comment for consideration with the null model, especially regarding its use at high sequence identities. The null model that the authors use calculates the expected number of cumulative identical fragments if all nucleotide differences between two genomes are attributed to mutations. However, if it is true that "when we compared one reference genome against representative genomes of all available genomovars, this fraction often approached 80% or higher when all recombination events detected with all possible partners in the analysis were summed", this means that when observing a pair of genomes, much of the SNPs that are observed likely comes from one genome recombining with a third genome (diversifying recombination). Therefore, ignoring this type of recombination in the null model can lead to a significant underestimate of the identical fragments in the null model: imagine a pair of genomes with 99.5% ANI, under an extreme case where 10% of the genome is under diversifying recombination which attributes to all the 0.5% difference in nucleotide identity. Under this case, 90% of the pairwise genome alignment should consist of identical fragments due to vertical inheritance. However, using the null model one should expect only approximately 16% of 1000 bp windows to be >99.8% ANI (This is calculated by asking if we have a Poisson distribution with a mean of $0.5\% \times 1000\text{bp} = 5$ SNPs per 1000 bp window, how many 1000 bp windows do we expect to have 0,1,2 SNPs in total; for the author's inquiry about how we got to 0.2% possibility of having 1000 bp windows being >99.8% ANI when overall ANI is 99%, it is calculated in the same way. I suggest the authors at least correct their pairwise ANI for diversifying recombination before applying the null model. I also suggest to the authors to mention the null model earlier in the manuscript instead of at the end (Figure 5) for clarity to the reader.

I thank the authors for the alteration and adding the citations to the Fraser et al. 2007 as well as Shapiro and Polz (2015), However I am not sure the Fraser paper was properly recognized since the text citing it didn't change and still says it focuses on recombination while the paper clearly pointed out recombination was not enough. Also, if I am understanding correctly, the authors put a strong emphasis on recombination being unbiased across the genome as a selling point, while in many modeling papers this is usually the assumption. Maybe the authors can emphasize it a bit more to say that while it is often taken for granted in models, there isn't a lot of empirical evidence?

REVIEWER COMMENTS

Reviewer #1 (Remarks to the Author):

Dear authors,

I read your MS with great pleasure in interest. Upon reading my comments below, it should be noted that I am not an expert on any microbial species in particular (esp not *Sal. ruber*). As a modeller, I consider my bioinformatic skills to be above the basic level, but not more so. Please keep that in mind when reading my comments.

In this review, you address an important problem: given the frequent exchange that is being shown by many recent papers (including your own), what happens to the microbial species concept, and which lines can be drawn? You discuss this in light of two opposing pressures: diversifying (point) mutations and cohesifying (is that a word?) recombination. You show that there are very strong cohesifying effects. If I understood correctly, you are even inferring fairly recent HGT events between isolates from geographically distant sites, but that this mostly happens within genomovars/phylotypes. This would imply your species (or microbes in general) are a) frequently exchanged between habitats, and b) that within this habitat they only recombine with their own genomovar/phylotype. The latter could be due to having specific niches (a proximity effect), or more farfetched reasons such as biased DNA uptake (seems to need a lot more assumptions). I find this concept a remarkable and novel insight, which although previously expected was never shown clearly. Although as you note you are also limited in scope by your data, you have set the stage for exciting new studies.

>Thank you for your comments and encouragement. You have captured the key methodologies and findings precisely above!

Here are a few points where I got stuck:

> I perhaps do not agree entirely on your focus on identifying "the real mechanism" behind these diversity patterns, as I'd rather see a good description of the data than read another discussion on what a species even is. But that is a philosophical discussion that is not really necessary here.

>The reviewer has a valid point above; we decided to describe the species issue in order to appeal to the general audience that sees it likely first time. The reviewer is right that if we were targeting a more focused group of readers (e.g., microbiologists), we could eliminate this part. We think, however, that it is better to keep the review of, and implications for, the species issues in order for the article to be accessible to a broader audience. Hence, we have not taken further action to respond to this comment.

> You refer to HGT being mostly driving by HR on line 118. I agree that this is a likely mechanism, but some MGEs seem to move in ways that may look like HR. E.g. some TEs can excise and leave behind their original target site, such that a new TE will quickly take its

place. I'm not sure it is necessary to discuss this in detail, but perhaps a brief mention on the possibility that this is driven by MGEs is good

>We mentioned that non-homologous recombination (mediated by MGE) later, in the Discussion section of paper, and we have modified a key sentence of the Introduction section to refer the reader there as well as to mention the possibility raised by the reviewer above. We have not examined carefully MGEs in this study; thus, we are unable to say more about them for now such as how frequent they are or what signatures they leave in the genome. This is probably a good topic for a follow-up study. That said, it is much more likely that non-homologous recombination serves as a diversifying force rather than a cohesive force, in general, in our view.

> In line 122-124 you write "a previous study indicated". As I didn't take the time to actually follow up on the reference (most readers won't, so let me play an actual reader of your paper), please elaborate what you mean here. Is it that you infer X amount of genomovars by extrapolating from the current rate at which you could infer donors? I am confused.

>This is edited for clarity and additional details were included from our previous study.

> Line 139: I really like this distinction between recombination resulting in cohesiveness vs. divergence, and I think it deserve a better priming in the introduction (its present already, but its limited).

>This is a good suggestion, and we have adopted it.

Do you think your observation of HGT being mostly a cohesive force is related to HGT acting as a repair for Muller's Ratchet / gene maintenance mechanism?

> yes, this could be the case, but we cannot think of how we could gather direct evidence in support (or not) of this hypothesis. If the reviewer has any suggestion, we would be happy to hear about it. We do mention this possibility/hypothesis in the revised Discussion section now to hopefully stimulate others (and ourselves!) to look into this hypothesis in the future.

> I would be careful in unpacking what is implied by local populations being relatively un-influential when it comes to recombination patterns. Some people use local to mean "both strains were isolated from the Black Sea", whereas others mean it came from the same dental cavity. Since the former describes millions of niches, and the latter describes a few at most, this difference matters for your interpretation. I think this part of the text (240-254) addresses this only partially, but should be carefully addressed.

>This is also a valid point, and we have edited the corresponding sentence to mean within the same niche.

> Why is Figure S5 not part of the main text? If you are limited by figures, could you make a simplified version of Figures S5 be part of the main text? This also relates to my last point

>Following the suggestion of the reviewer, we have made Figure S5 a main figure (now Figure 4).

> I think the last sentence of the *Sal. ruber* section deserved much more emphasis throughout the entire MS: "we recommend using ANI value distributions". While you are probably not the first to say this, you are one of the first to highlight how important it is in this particular metric. In the age of "big data", I am perplexed at how many single-value things are being measured to this day, even when we know how misleading they can be. So my general advice would be to let this message shine a bit more in the abstract/introduction, although that is of course ultimately your own choice.

>This is also a good suggestion too, and we have adopted it.

Minor points:

> Please change the colours for Figure 1 and 2 to be colour-blind friendly.

>We have played with the colors of this figure a lot in the past, including to follow the suggestion of a color-blinded member of our lab and we were not able to improve it further. The figure has a lot of useful information that is not possible to reflect well with only 2-3 colors or a single color and its gradient. When we did the latter, the non-color-blinded people complained about the figure. So, we have decided to not do further changes to the figure.

> Figure 4 is a bit crowded. Perhaps add some bigger margins because there is a lot going on.

>Done, including to increase the fonts (now Figure 5).

Reviewer #2 (Remarks to the Author):

I agree with the authors that it is imperative to understand what mechanisms underlie the creation and maintenance of sequence-discrete species and genomovars. However, I found the exact hypothesis and methods of this study poorly explained in the (long) introduction. The discussion on clones, strains, genomovars, species is quite involved and could benefit from a Figure depicting the ANI cut-offs between these classes proposed by the authors. The authors state that both ecological differentiation and recombination influence divergence between sequence clusters which is very true, but I do not recognise a clear research question that explains how to distinguish between these forces. What does the sentence "recombination frequency coupled to ecological cohesiveness among members of these units might account for the species and intra-species clusters observed previously." actually mean? I also disagree with some of the assumptions in the intro, see comments below.

A lot of the text in the Results seems Introduction to me. Also, the lack of any subheaders makes it more difficult to grasp the main thrust of the argument. There does not seem to be a lot by way of data when reading the text (e.g. "Therefore, it appears that these *Sal. ruber* genomes are engaging in genome-wide, rapid recombination" does not seem to be

supported by any statistics/formal analysis). The Figures show some interesting data but since many researchers have done comparative genome analyses (esp on E coli), what is the added insight presented here exactly? For instance, the fact that genetic exchange is frequent, and also that it is more common between more closely related isolates is convincingly shown here, but this is not exactly a novel result. The high frequency of recombination indicates that it is a force for cohesiveness, but any evidence (dis)proving 'ecological selection' as a force of cohesiveness/divergence (requiring detailed, hard to collect data from the field) is much weaker. The authors state "instead of attributing the sequence-discrete units to either ecological or genetic (e.g., recombination) mechanisms, our model suggests that the two types of mechanisms may operate together, which represents a departure from previous models of speciation" but I just do not see this supported by the data. It is not explained why the focus is on the two chosen species. Demonstrating a general pattern requires a range of species but an indepth species-level analysis is also useful, however, inclusion of a second species is almost a distraction. The results introduce the concept 'phylogroup' which is yet another subdivision and one that was not even mentioned in the introduction, which I found confusing.

>We disagree that our Introduction is not clear and the questions we pursued are not outlined clearly and comprehensively. Our other two reviewers do not seem to share the concern of this reviewer either by statements like this (paper) was exciting to read and novel etc.

Indeed, the frequency of recombination has been shown to be higher than point mutation by the studies mentioned below but whether or not this could lead to cohesion is a different question, or -at least- the frequency alone is not enough for cohesion when it occurs in only a few regions of the genome (e.g. genomic islands). Cohesion requires to show that recombination is random across the genome i.e., it occurs everywhere. This has been missed by the studies mentioned below by this reviewer or reviewer #3, as this is also clearly explained in the Introduction of our paper. Further, previous studies have accessed historic recombination using methods that have important assumptions that are very likely violated by the data such as lack of selection for the HGT events detected (and thus, their detected frequency). Instead, here we focused on recent recombination and no assumptions, which we believe is a more robust and especially transparent way to investigate the role of recombination. This was clearly implied in our previous text, but we also made several edits to more clearly reflect the points mentioned above in our revised paper.

L4

"Recent large-scale surveys of prokaryotic communities (metagenomes) as well as isolate genomes have revealed that their diversity is predominantly organized in sequence-discrete clusters or units that may be equated to species. Specifically, genomes of the same species commonly show average nucleotide identity (ANI) of shared genes >95% between them and ANI <85% to members of other species. Intermediate identity genotypes, for example, sharing 85–95% ANI, when present, are generally ecologically

differentiated and scarcer”.

Is his not circular reasoning? Sequence clusters here are defined according to species status, rather than the other way around. Also, what is ‘ecologically differentiated’ based on? (gene content? Presence/absence across samples? Fitness measurements across different conditions?)

>We do not agree that it is circular logic; neither is correct that the clusters are based on species status. Rather, our previous analysis (e.g. Jain et al., Nat. Comms. 2018 or more recently for the genomovar gap, Rodriguez-R et al., mBio 2024,) did not consider the species status, and only overlaid the species information at the very end, after the clusters appear during our analysis, in order to see how consistent this species information is or not with the ANI peaks and valleys.

We have provided a clear explanation in the manuscript that ecological cohesiveness refers to similar ecology, that is, similar abundances under the various growth conditions. Perhaps our reviewer missed that part? This explanation is also provided clearly in the abstract in the next sentence i.e. “Ecological cohesiveness was inferred based on higher similarity in temporal abundance patterns of genomes of the same vs. different units “. And yes, in general higher ecological cohesiveness comes with higher gene content similarity as our previous work has recently shown (e.g., Rodriguez-R et. al., mBio 2024). We believe all this is clear enough and thus have made only minor edits to make the text hopefully a bit clearer.

L43

“Notably, given an estimated mutation rate of $\sim 4 \times 10^{-10}$ per nucleotide per generation and between 100 to 300 generations per year (2014), it would take two distinct lineages of a gut microbe such as Escherichia coli at least 100,000 years since their last common ancestor to accumulate 0.5% difference (i.e., fixed mutations) in their core genes or 99.5% ANI”

But SNPs used in ANI calculations are also introduced via homologous recombination, which is around 3.6 times as frequent (Torrance et al 2024 PNAS) so this calculation is not correct.

>We made it clear in the revised manuscript that the estimation above is based on the assumption of no recombination. Also, homologous recombination will more often purge mutations rather than create new ones, so if one considers recombination, then the evolutionary time is likely longer, not shorter, due to this.

L46

“Therefore, given enough time, it is possible to have ecological purging of diversity even at around the 99.5% ANI level (let alone at 95% ANI) that accounts for the ANI patterns observed.”

I assume ‘ecological purging’ here equates to purifying selection. Why would all SNPs be purged though? Not all will be very deleterious.

>No. Purging refers to loss of diversity due to competitive exclusion acting at the whole cell/organism level (diversity purging) not to purifying selection acting at the sequence level, and we made it clearer in the revised text.

L57

“[Note that this is a fundamentally different mechanism than the sexual reproduction in eukaryotes and the accompanying biological species concept, although the ultimate outcome in terms of species cohesion may be similar. That is, gene exchange does not occur during a meiosis step but via vectors of horizontal gene transfer mediated by recombination. Hence, we opted to not use biological or sexual species here and, instead, refer to them as recombinogenic species].”

Of course meiosis is very different from transduction/transformation/conjugation but genetic exchange in both eukaryotes and prokaryotes DOES act as a cohesive force, as the authors note themselves, so I do not understand this sidenote. (I also note that the later “Our results also suggest that bacteria, and likely other microbes, may evolve more “sexually” than previous thought.” seems to obviate this view.)

>It does not seem like we disagree here with the reviewer on this point. We made the differences clear between eukaryotic vs .prokaryotic recombination for the general audience and young students in the field that may not be very familiar with this topic. The sentence he/she quotes at the end of the comment above targets the general audience that may not know much about recombination in prokaryotes. Hence, we have not edited it further.

L62

“indeed, several studies have concluded that the frequency of homologous recombination could be high enough to have a greater effect on sequence evolution than point (or diversifying) mutation, as has been shown to be the case for Campylobacter species and other taxa more recently.”

There are both older (more original) and newer (better) studies available that clearly demonstrate this point (which the authors do not take into account in their divergence time calculation above). Indeed, it is the norm, as Torrance et al show “For most species (82%), the homologous recombination rate was found to be more impactful on genomic evolution than mutations alone ($r/m > 1$).”

>Please see our response above about spatial biases (in homologous recombination) across the genome.

L65

“However, these studies have not been able to assess whether recombination is occurring across the genome (that is, it is not biased spatially and/or functionally) to serve as a force of cohesion or/and to what extent it accounts for the sequence-discrete units.”

The Torrance et al study is based on WGS data so this statement is not correct.

> As explained above, this study and other mentioned below have not shown that recombination is unbiased across the genome or directly link recombination to the ANI clusters of diversity such as the genomovar level and the ecological relatedness of the genomes studied as performed here. Hence, we respectfully disagree on the statement above and have taken no further action to respond to this comment.

L67

Regarding the general point “recombination is unlikely to lead to species cohesion in such cases since the non-recombining segments of the genome will continue to diverge” and genomic islands: GIs rapidly change in gene content and the point about ANI divergence does not apply here, the core genome will be reflective strain/species divergence, which at least in *C. jejuni* is affected by a high rate of HR (see again the Torrance reference).

>We are unable to follow this point as it is not very clear to us what the reviewer is trying to say. “Non-recombining segments of the genome” used in our text refers to both shared (core) parts and genomic islands of the genome, not only genomic islands as it was obvious from the preceding text.

L167

“mutations created by point mutation within the same time (m),” does this method take into account SNPs introduced by HR?

>No, and we have made this clear.

L291

biased functions?

>Biases IN functions. Corrected. Thank you for catching this.

L310

The authors cannot just state “In fact, our recent work suggests that similar patterns of recent gene exchange can be observed in human and bacterial viruses” without any explanation/evidence.

>We have added our recently accepted paper that backs up this statement (Aldeguer et al., mBio 2024).

Abstract

I think the Abstract is very unclearly phrased. It ends with the authors stating that they have found the answer for what a species is, but what this answer is, remains very hazy. Also, ‘persistent’: what does this mean in this context? Detected in chronological sequencing studies? ‘a novel methodology to identify recent recombination events’: this needs a little bit of detail, what is novel about it? (this point is not explained in the Introduction either) ‘unbiased (i.e., not selection driven) horizontal gene flow’: I take issue with this, as HGT itself is followed by selection rather than is driven by it (unless the authors mean that there is second-order selection for HGT, which is not obvious here). ‘diversifying mutation’ is not a term in general use so please just use ‘point mutation’

>HGT events could be selected for and hence, their frequency will increase with higher the selective advantage of the transferred gene; or they do not confer a selective advantage and thus, they could stay in the genome or be removed. The latter cases are probably more frequent than the former, but the former is what we see more often in the sequenced genomes because they get fixed in the genome (as opposed to being lost over time). So,

selection could affect what HGT vents are being detected and counted (e.g. against point mutation), and thus could offer biased results e.g. count more the latter than the former events. It is not clear enough to us what point the reviewer is trying to make above. We have clearly defined diversifying mutation in the sentence below, and this term is indeed used sometimes. So, we believe that no additional change is needed in this sentence: “Indeed, several studies have concluded that the frequency of homologous recombination could be high enough to have a greater effect on sequence evolution than point (or diversifying) mutation”.
Unfortunately, we do not have room in the abstract to explain more the terms used (we are already a bit above the limit of the journal guidelines).

Reviewer #3 (Remarks to the Author):

Conrad, Brink et al, present a model that ecological cohesiveness coupled with high rates of recombination are responsible for sequence discrete units. Their work largely focuses on the analysis of their model organism, *Salinibacter ruber*, a common community member found in salterns. They apply their findings to a genomic collection of *Escherichia coli* isolates to ascertain if their observations are generally applicable to other bacteria. More specifically, they analyse their genomic collections to observe the role of unbiased homologous recombination and its frequency. In the latter half of the results and discussion they hypothesize what their observations could mean to their current working model of the emergence and maintenance of sequence discrete units. The discussion of what is a species in the context of bacteria or archaea is always intriguing and not, unsurprisingly, without a deal of hypothesizing as is the case in this manuscript. Some clarity or modifications to the methods and expansion of the discussion would improve the manuscript. Below are some comments which I hope the authors will find useful.

>Thank you for your time and comments. We have made several edits and additions to make our underlying methodology clearer.

In this study the authors discuss recombination in genomovars and phylogroups. The definition of a phylogroup for *Salinibacter ruber* is given by a single line in the methods (line 320-321). How the phylogroups were defined should be more explicitly stated.

>We have modified the corresponding sentence to be clearer.

Moreover, as figure 1 shows the phylogroups for both *Sal. ruber* and *E. coli* are not of equal size and diversity. Lines 137-153 discusses recombinant genes within and between different phylogroups. Here the size of the phylogroup should be taken into consideration as a larger phylogroup would mean more pairwise intra-phylogroup and less to inter-phylogroups comparisons compared to less populated phylogroups.

>True, but we have made a strong effort to normalize for the amount of diversity (measured by ANI in our case) as well as the number of genomes for each group compared. See for

example our response below about Figure 4 (Figure 5 in the revised manuscript) but also the second paragraph of the Methods section that provides all details.

In the definition of genomovars the authors refer to this as 99.5%-ANI (line 21) I know this is the midpoint, however at line 108 genomvar is given as ANI >99.5% and yet more values are given in figure 1 for genomvars. It is a small point but being more consistent would greatly aid the reader.

>We don't think our values are inconsistent. Genomovar gap is between 99.2 and 99.8% ANI usually, and the specific genomovars of *Sal. ruber* typically show >99.5% ANI within the genomovar. Figure 1 is not coloring genomovars, it just shows the range in ANI values observed, colored differently in order to be easy to see the common ANI gap at the genomovar level (and there a couple exception to the 99.2-99.8% gap as our recent paper also showed for other microbial species, Rodriguez-R et al., mBio, 2024). Accordingly, we have taken no further action to respond to this comment.

The cutoff of using 99.8% ANI as the recombination threshold needs more justification. First, the authors have not provided a rationale for selecting 99.8% as the recombination cutoff (line 141). It appears this threshold might have been chosen because 99.8% represents the upper limit for defining a genomovar. However, the manuscript neither mentions this assumption nor justifies it.

>The choice of 99.8% was mostly to account for 1-2 SNPs that represent sequencing errors and are likely found in the genomes, especially the high-draft ones. And, to look only at recent events, not historic recombination that is challenging to detect and quantify due to sequence amelioration. We have done our analysis with a 100% threshold as mentioned in the paper and the latter did not affect conclusions. We made sure this is clearly mentioned in the revised text. This cut-off (99.8%) is not linked to the ANI gap although it helps our analysis that the ANI gap is below this level (e.g. enough signal over background ANI for between genomovar or phylogroup comparisons).

Second, the authors do not present an underlying null model in most of the recombination rate/fraction estimations. While using identical regions as indicators of recombination is common, this approach is typically applied to genomes from different species, where the genetic distance is sufficiently large, making the likelihood of identical regions arising from vertical inheritance very small (references 1, 2 below). However, since the authors are comparing genomes within the same phylogroup versus those from different phylogroups—and considering that genomes within the same phylogroup can exhibit ANI values greater than 98.5%—it's crucial to consider how genes with ANI greater than 99.8% could be shared between two genomes by vertical inheritance alone. For instance, a 1000 bp gene at 99% ANI has a 0.2% chance of being >99.8% ANI due to vertical inheritance, not accounting for the fact that some genes are more conserved due to negative selection.

>it is not clear to us where the 0.2% comes. Could you please add an explanation or a reference for it?

Please see also our response below about our modeling analysis that estimates the total genes in the genome that should be expected to be identical among genomes in the absence of any recombination as a function of their ANI relatedness.

A few places where comparing against a null model that includes only vertical inheritance (and ideally considers negative selection) could improve the analysis: A, Comparing the r/m ratio of genomes within a phylogroup versus between phylogroups (lines 172-173, 209-210): Here, it is essential to include a null model because genomes within each phylogroup have higher ANI than those in different phylogroups. Thus, even with complete vertical inheritance, a larger genome fraction is expected to exceed 99.8% ANI, which could be mistakenly identified as recombination. B, calculating overall recombination (sum of 1 vs. all possible partners) for every genome (Lines 189-190): In this case, the estimated amount of recombination between genome pairs is summed across ~200 genomes. Therefore, even if vertical inheritance only leads to 0.1% of the genome being >99.8% ANI, this could falsely identify 20% of the genome as recombined.

> In the quantitative estimates of recombination shown in figure 4 (Figure 5 in the revised manuscript) and elsewhere, we have taken into account the ANI within the groups being compared and normalized for it. For instance, note the empty circles on figure 4 that represent the number of genes that should be found to be >99.8% nt identity based on the ANI of the genome compared and assuming no recombination. Note how these values are decreasing with lower ANI values of the genomes compared, consistent with the expectation and what the reviewer mentioned above. This is exactly the null model that the reviewer is calling for. We provide the details of this model in the Methods section (2nd paragraph is devoted to it) as well.

Further, for the number of genes of the genome found to recombine in figure 4 (Figure 5 in the revised manuscript) and elsewhere, please note that this is unique, non-redundant genes. That is, if a core gene is 99.8%-100% across genomes due to sequence conservation (such as the 16S rRNA gene) or recombination, this is counted only once, not as many times as the number of genomes found in figure 4 and elsewhere. Hence, our estimates do not overcall/inflate recombination, and this caveat/limitation mentioned by the reviewer “leads to 0.1% of the genome being >99.8% ANI, this could falsely identify 20% of the genome as recombined” is effectively taken into account and does not affect our conclusions.

The authors emphasize (i.e. abstract, conclusion section) that their model suggests that ecological or genetic mechanisms operate together to form sequence discrete units, and the model is a departure from previous models that emphasize one or another. This might be a little unfair to others, since the previous models cited by the authors clearly state that recombination alone is not sufficient for forming stable clusters that could lead to speciation (reference 3 below), or that the gradual decline of recombination with divergence is an important step in completing the ecological speciation process (reference 4 below), it fact there appears the author’s model has similarity to what is proposed

ecological speciation in (reference 4 below). Therefore, perhaps an expansion of the discussion to include previous works would be warranted.

>We agree, and we have made extensive edits to the Discussion section to better present previous literature, and specifically cite appropriately the important prior work of Fraser et al. 2007 as well as the excellent review of Shapiro and Polz (2015). We also added text to make it even clearer how our work differs from these and other previous studies. That is, the model presented here has novel elements such as it directly links recombination frequency to the ANI clusters, rigorously shows the (lack of) spatial bias in recombination across the genome, and considers the role of ecological relatedness at the genomovar level within species, which we believe is unprecedented. There are common elements with previous studies, as the reviewer also points out above, but the models of how exactly this work are not identical, and we tried to better explain the key differences in the revised text.

1. Smillie CS, Smith MB, Friedman J, Cordero OX, David LA, Alm EJ. Ecology drives a global network of gene exchange connecting the human microbiome. *Nature* 2011; 480: 241–244.
2. Sheinman M, Arkhipova K, Arndt PF, Dutilh BE, Hermsen R, Massip F. Identical sequences found in distant genomes reveal frequent horizontal transfer across the bacterial domain. *eLife* 2021; 10: e62719.
3. Fraser C, Hanage WP, Spratt BG. Recombination and the Nature of Bacterial Speciation. *Science* 2007; 315: 476–480.
4. Shapiro BJ, Polz MF. Microbial Speciation. *Cold Spring Harb Perspect Biol* 2015; 7: a018143.

REVIEWER COMMENTS

Reviewer #1 (Remarks to the Author):

I am happy with the current state of the manuscript.

>thank you for your (previous) suggestions, time and encouragement.

Reviewer #2 (Remarks to the Author):

I was surprised to see the positive assessments of the other two reviewers (and editor) after the first round of review.

Just looking at the revised Abstract (after all the section of the paper highlighting why the paper is novel and worthwhile):

“Ecological cohesiveness was inferred based on higher similarity in temporal abundance patterns of genomes of the same vs. different units, while recombination frequency was shown to be genome-wide and to have two times or greater impact on sequence evolution than point (diversifying) mutation.”

If the authors indeed would demonstrate a rise or fall in frequency of related sequence types over time, this would indeed have been a very interesting demonstration of ecological cohesiveness. However, none of the figures show this result! (Indeed, the term ‘temporal’ is not even mentioned anywhere else in the paper – it seems that is based on anecdotal evidence only – not sure where in the long paper).

> S8 is modified from a previous Nat. Comm. publication (Viver et al., 2024) and thus, we do not want to have it as a main figure. Indeed, readers that are more interested in the ecological aspect of the genomes analyzed here would have to read our previous, related publication, and we believe this is ok since an adequate summary of this and other relevant publications was already included in the manuscript. Time-series metagenomics and isolates are mentioned clearly in our Results section for *Salinibacter ruber* as well.

That recombination is a force of cohesiveness is well-known and its impact has been shown to be even greater in other taxa. Contrary to what the authors assert, other researchers have taken a genome-wide approach to detecting homologous recombination rate, e.g. see <https://doi.org/10.1093/gbe/evs043> (published 12 years ago!). (‘recombination frequency was genome-wide’ is not correctly phrased if meant to mean that recombination frequency was similar across the genome btw.)

> That paper is only about *Helicobacter* (so limited in scope) and in fact says that the authors found several genes with no recombination. This is enough for diversifying

mutation to increase diversity in these genes, and thus lead to diversification, despite any cohesive action due to recombination. Importantly, that paper has not linked recombination to the ANI clusters, which is an important limitation of most other related papers too. Our study addressed these exact issues. Thus, we do not feel we need to cite more similar studies or add more discussion on the topic.

Why the authors feel that they can continue on with the statement “therefore, our results represent a departure compared to previous models of microbial speciation that invoke either ecology or selection-driven recombination, but not their synergistic effect.” is completely unclear to me. Many have already argued that ecological cohesiveness and recombinational cohesiveness go hand in hand: for instance, by occupying the same spatial niche, there is a higher chance of cell-cell (or cell-eDNA) contact between different intra-species clusters. (Also, ‘diversifying mutation’ is not generally used as a term (or used at all) and is confusing as any mutation results in new diversity (homoplasies aside).)

>Correct but no study has offered unequivocal experimental data in support this model/hypothesis and there are several folks out there that still believe that bacteria evolve primarily asexually with infrequent HGT driven by selection. Hence, the matter is far from resolved. What you wrote above remains without experimental validation, especially in terms of showing that recombination is unbiased -neutral- across the genome, until our study. Please keep also in mind that the ecological species concept and the recombination/sexual species concept, the two most dominant theories to explain microbial speciation, do not necessary invoke one another for explaining speciation (like our study does).

This is just the Abstract. I have no desire to revisit all the other arguments that the authors have not bothered to properly address.

>Thank you very much for your time, regardless.

Reviewer #3 (Remarks to the Author):

Thanks to the authors for the revised manuscript and response to the queries and comments. There are two points to the authors. One which hopefully adds clarity to the original review comments and the other just a textual recognition of previous work.

Apologies for not being clear. The null model, sorry about missing out on the null model part. However, I still have a comment for consideration with the null model, especially regarding its use at high sequence identities. The null model that the authors use calculates the expected number of cumulative identical fragments if all nucleotide differences between two genomes are attributed to mutations. However, if it is true that “when we compared one reference genome against representative genomes of all available genomovars, this fraction often approached 80% or higher when all

recombination events detected with all possible partners in the analysis were summed”, this means that when observing a pair of genomes, much of the SNPs that are observed likely comes from one genome recombining with a third genome (diversifying recombination). Therefore, ignoring this type of recombination in the null model can lead to a significant underestimate of the identical fragments in the null model: imagine a pair of genomes with 99.5% ANI, under an extreme case where 10% of the genome is under diversifying recombination which attributes to all the 0.5% difference in nucleotide identity. Under this case, 90% of the pairwise genome alignment should consist of identical fragments due to vertical inheritance. However, using the null model one should expect only approximately 16% of 1000 bp windows to be >99.8% ANI (This is calculated by asking if we have a Poisson distribution with a mean of $0.5\% \times 1000\text{bp} = 5$ SNPs per 1000 bp window, how many 1000 bp windows do we expect to have 0,1,2 SNPs in total; for the author’s inquiry about how we got to 0.2% possibility of having 1000 bp windows being >99.8% ANI when overall ANI is 99%, it is calculated in the same way. I suggest the authors at least correct their pairwise ANI for diversifying recombination before applying the null model. I also suggest to the authors to mention the null model earlier in the manuscript instead of at the end (Figure 5) for clarity to the reader.

> The null model assumes no recombination and we have assessed the impact of diversifying recombination elsewhere in the manuscript. What the reviewer suggests, that is to account for diversifying recombination in normalizing ANI towards getting an ANI closer to the actual ANI if there was no recombination, is NOT possible for several reasons such as that we do not have all the partners/donors (e.g. we are under sampling the total strains present) and, even if we had all partners, it is not clear to us what the usefulness of normalized ANI values would be. We mean that the key point of the null model was to provide a qualitative reference only, as we don’t perform any statistical test against it to accept or reject a null model. It is simply a baseline model of what the fraction of identical genes would be like with only random point mutations added between two genomes to yield a target ANI value. Even if we were able to incorporate diversifying recombination into the ANI values, which is not possible as explained above, that addition would likely decrease the ANI values, and thus identical segments/genes rather than increase them as -we think- the reviewer implies.

I thank the authors for the alteration and adding the citations to the Fraser et al. 2007 as well as Shapiro and Polz (2015), However I am not sure the Fraser paper was properly recognized since the text citing it didn’t change and still says it focuses on recombination while the paper clearly pointed out recombination was not enough.

> We believe that the reviewer has misinterpreted the main message of this paper (or please let us know if we have misinterpreted your comment below). That paper mostly says that recombination can be the force of cohesion not the opposite, which is what the reviewer implies, if we have understood correctly their comment. Perhaps the reviewer is confused (?) by this paragraph of the Fraser et al., paper, toward its end (?)

From Fraser et al.

If recombination is more frequent than this, a threshold is crossed and recombination starts to act as a cohesive force on the population by breaking linkage between alleles and reducing genetic clustering. Such a situation could in principle lead to dynamic speciation by chance drift, **but only if the amount of variation within the population is sufficient for recombination rates to vary appreciably between members of the population. On the basis of current estimates for the species we have studied, this does not occur, but it should not be ruled out.** Thus, in general, bacteria can and do form sexual species, and mechanisms involving allopatry or niche specialization must be invoked in speciation. In this case, the situation is largely analogous to speciation in higher organisms, without the complications associated with sexual mating choice

The way we interpret this paragraph is that it refers to speciation from within the population by chance alone or under neutral (not selection-driven) conditions, i.e., a subpopulation with higher intra-subpopulation recombination compared to other subpopulations of the total population is needed for this, and the scientific community was not able to document such cases at the time of the writing of this review, which is 15 years ago. Indeed, our study provides a way to do this analysis and shows that, at the phylogroup level within species, this differential frequency of recombination does occur!

Also, if I am understanding correctly, the authors put a strong emphasis on recombination being unbiased across the genome as a selling point, while in many modeling papers this is usually the assumption. Maybe the authors can emphasize it a bit more to say that while it is often taken for granted in models, there isn't a lot of empirical evidence?

>This is a good point, and we have adopted it. We had some text that was saying exactly this previously, but we expanded on that text in the current, revised version.